# Advances in Synthesis and Ignition Performance of Ionic Liquid–Hydrogen Peroxide Green Propellants

**DOI:** 10.3390/molecules30081789

**Published:** 2025-04-16

**Authors:** Yongting Zhang, Xing Zhang, Dangyue Yin, Qinghua Zhang

**Affiliations:** 1School of Astronautics, Northwestern Polytechnical University, Xi’an 710065, China; 18091925902@163.com (Y.Z.); zhangxing@nwpu.edu.cn (X.Z.); qinghuazhang@nwpu.edu.cn (Q.Z.); 2National Key Laboratory of Solid Propulsion, Xi’an 710065, China

**Keywords:** liquid propellant, ionic liquids, hydrogen peroxide, ignition delay

## Abstract

The ionic liquid–hydrogen peroxide propellant system has emerged as a promising green propellant candidate, synergistically combining the unique advantages of ionic liquids (such as negligible vapor pressure, low melting points, high thermal stability and structural tunability) with the merits of hydrogen peroxide (including high density, low volatility, minimal viscosity, reduced corrosivity, and environmentally benign decomposition products). In this work, we provide a comprehensive review of the synthesis strategies and ignition performance of the ionic liquid–hydrogen peroxide propellant system, systematically categorizing them into two classes: “self-igniting propellants” and “promoter-dependent propellants”. This review emphasizes the critical role of anion-specific design and catalytic engineering in advancing the performance of ionic liquid–hydrogen peroxide propellant systems, while also addressing the current challenges and future directions in this rapidly evolving field.

## 1. Introduction

Propellants, defined as energetic materials that undergo rapid combustion or decomposition to generate high-pressure gases for thrust production, play a pivotal role in aerospace propulsion, satellite trajectory control, and gas generation systems [1]. While solid propellants dominate certain applications, liquid propellants offer superior thrust modulation, enhanced combustion reliability, and restart capability, making them indispensable for advanced rocket engines and spacecraft [2,3]. Liquid propellants can be categorized into two types: monopropellants and bipropellants. Unlike monopropellants, bipropellants do not require an external ignition device, which simplifies the engine structure, reduces costs, enhances combustion reliability, and allows for multiple restarts and shutdowns to improve flexibility [4].

The bipropellant consists of two components: an oxidizer and a fuel. The oxidizers are usually strong oxidizing compounds such as nitric acid, dinitrogen tetroxide, white fuming nitric acid, and red nitric acid [5]. These substances are not only highly corrosive and volatile but also produce significant amounts of nitrogen oxides upon combustion [6], posing severe toxicity risks to human health and causing substantial environmental pollution. Hydrazine and its derivatives exhibit excellent performance as propellant fuel and have widespread applications in liquid propellant systems, but they are explosive and highly toxic chemicals with a low boiling point and strong volatility. The use of hydrazine compounds will cause many destructive consequences to human health and ecological environments, and in practical applications, the transportation cost of such substances is high, and the storage is difficult [7,8,9,10].

This critical safety–environmental paradox has driven intensive research toward green liquid propellants. Ionic liquids, a class of salt compounds composed of organic cations and organic or inorganic anions usually with a melting point below 100 °C, stand out for their low toxicity, non-volatility, short ignition delay, structural tunability, and high safety [11,12]. Hydrogen peroxide (H_2_O_2_) is a common industrial chemical and environmentally friendly oxidizer which is produced in extremely large quantities. It has many advantages of low toxicity, low vapor pressure, and non-toxic decomposition products [13,14]. In addition, there have been significant improvements in its stability and storability [15]. Therefore, liquid propellants based on the combination of ionic liquid–hydrogen peroxide are expected to realize the true greening of self-ignition liquid propellants. Therefore, developing such propellants is crucial for promoting the green process of space power in the future.

In fact, only a limited number of existing energetic ionic liquids, primarily those incorporating borohydride or thiocyanate anions, exhibit spontaneous hypergolic ignition with high concentration of H_2_O_2_. For ionic liquids that do not exhibit this capability, catalytic promoters, such as iodine or copper-based organometallics, have been strategically employed to achieve millisecond-scale ignition delays [16]. This review systematically categorizes ionic liquid–hydrogen peroxide propellant systems into two distinct classes, “self-igniting propellants” and “promoter-dependent propellants”, with a focus on four anionic families: borohydrides, thiocyanates, dicyanamide, and other emerging anions. We critically summarized their synthesis pathways, structure–ignition performance relationships, and catalytic effects, aiming to establish development ideas and explore application directions for next-generation green propellants.

## 2. Ionic Liquid–Hydrogen Peroxide Liquid Propellant Capable of Self-Ignition

### 2.1. ILs Based on Borohydride-Rich Anions

Among the existing ionic liquids, only a small amount of ionic liquids can directly and rapidly self-ignite with a high concentration of H_2_O_2_, which usually have borohydride bonds with strong reducing ability.

In 2011, Schneider et al. first proposed studying the spontaneous combustion behavior of ionic liquids in H_2_O_2_ [17]. They speculated that ionic liquids with metal hydride anions could show high solubility of H_2_O_2_ in ether and prepared ionic liquid borohydride with subsequent conversion to [Al(BH_4_)_4_]^−^ ionic liquids. Trihexyltetradecylphosphonium (THTDP) chloride was utilized to enable facile, quantitative anion exchange of Cl^−^ for BH_4_^−^. The resulting new material [THTDP][BH_4_] (IL-1) was a viscous ionic liquid. Subsequently, another ionic liquid [THTDP][AI(BH_4_)_4_] (IL-2) was synthesized by reacting with a slight excess of aluminum borohydride. The structures of IL-1 and IL-2 are shown in Figure 1. The two ionic liquids were then ignited with 90% and 98% H_2_O_2_, respectively, demonstrating their ability to undergo spontaneous combustion with H_2_O_2_ without a catalytic promoter. Remarkably, IL-2 exhibits fast self-ignition performance with both 90% and 98% H_2_O_2_, with ignition delay times (IDTs) less than 30 ms, while the IDT of IL-1 was longer than 3 s. These tests confirm that ILs based on borohydride-rich anions are universally reactive, offering a new approach for the designing of novel self-ignition ionic liquids with H_2_O_2_. However, the harsh synthesis conditions and poor water stability of these ionic liquids seriously restrict their extensive application as propellant fuels.

In 2017, Bhosale et al. conducted a theoretical investigation on the performance parameters of 14 ionic liquids (IL-3 to IL-16, Figure 2) with different oxidizers including H_2_O_2_, using the NASA-CEC-71 program [18]. As shown in Table 1, the effect of their performance parameters was analyzed in detail and compared with 1,1-dimethylhydrazine (UDMH). The results indicated that all ionic liquids exhibited superior performance with H_2_O_2_ compared to other oxidizers. Specifically, the ionic liquids containing borohydride-rich anions demonstrated higher density-specific impulse (ρI_sp_) values than UDMH. Among them, IL-3 exhibited the highest ρI_sp_ out of all the oxidizers as it had higher density than any other ionic liquid, achieving a ρI_sp_ of 319.8 g s cm^−3^ with H_2_O_2_. Therefore, combinations of ionic liquids based on borohydride-rich anions and H_2_O_2_ oxidizers can be recommended as future green propellants. However, since these experimental data were based on theoretical calculations and no actual drop tests were conducted, further verification of the spontaneity of these ionic liquids is necessary. Additionally, unless specifically noted, all IDT data presented in this study were obtained through experimental measurements.

In order to further study the above research achievements, Bhosale et al. synthesized several green hypergolic fuels with borohydride bond-rich anions, and investigated their hypergolic reactivity with 95% H_2_O_2_ [19]. As shown in Figure 2, the ionic liquids included [EMIM][BH_3_CN] (IL-3), [AEIM][BH_3_CN] (IL-4) and [EMIM][BH_4_] (IL-17), which were synthesized by simple metathesis reactions involving halide salts and metal borohydrides in selective solvents (CH_2_Cl_2_ or MeCN) at room temperature. IL-17 exhibited a short IDT of 18.5 ms with 95% H_2_O_2_, while IL-3 and IL-4 exhibited longer IDTs (>1000 ms) with 95% H_2_O_2_ in both fuel-rich and oxidizer-rich conditions. However, IL-3 and IL-4 were liquid, but IL-17 was solid at room temperature. Furthermore, the theoretical specific impulse (I_sp_) of ionic liquids was calculated with 95% H_2_O_2_ using NASA CEA 400 software under specified conditions, considering a range of oxidizer-to-fuel ratios (O/F, 0.5–5.5). IL-3 and IL-4 showed the lowest I_sp_ of 253 s at an O/F of 3.5 and had a fairly low viscosity (η of 17 and 19 mPa s), good thermal stability (T_d_ of 265 and 247 °C), and acceptable density (ρ of 0.95 and 0.98 g cm^−3^), suggesting that ILs based on borohydride-rich anions have application prospects as H_2_O_2_-based green spontaneous combustion propellants.

In 2023, Wang et al. synthesized four novel hypergolic fluids based on borohydride ionic liquids [1-ethyl-3-methylimidazolium borohydride ([EMIM][BH_4_]) or 1-butyl-3-methylimidazolium borohydride ([BMIM][BH_4_])] by an in situ synthetic method in an organic super-base [1,5-Diazabicyclo[4.3.0]-5-nonene (DBN) or 1,8-Diazabicyclo[5.4.0]undec-7-ene (DBU)], as shown in Figure 3 [20]. All these hypergolic fuels exhibited high densities and low viscosities due to the incorporation of super-bases with high densities and low viscosities, which show significantly superior properties compared to those of pure [BMIM][BH_4_] (ρ of 0.91 g cm^−3^ and η of 486.6 mPa s). Additionally, the hypergolic fluids with 90% H_2_O_2_ demonstrated an acceptable IDT, with a minimum of 28.3 ms, which is lower than that with WFNA. That is attributed to the formation of a homogeneous mixed layer and no occurrence of secondary rebounds during drop tests conducted with H_2_O_2_, indicating their strong potential as green fuels for use in green propellant systems. The organic super-base acted as an important solvent for the simple and low-cost synthesis of borohydride-containing ionic liquids; meanwhile, it is also the main component of fuels that possess high density and low viscosity, and the ionic liquids in DBN and DBU acted as triggers to contribute to the self-ignition of DBN and DBU with 90% H_2_O_2_.

The physicochemical properties of all ionic liquids based on borohydride-rich anions are shown in Table 2. To date, the anions of borohydride-based ionic liquids that can self-ignite mainly include [BH_4_]^−^, [BH_3_CN]^−^ and a singular example of [Al(BH_4_)_4_]^−^. Although theoretical calculations have demonstrated that the ionic liquids containing [BH_2_CNBH_3_CN]^−^ and [PH_2_(BH_3_)_2_]^−^ also exhibit high performances with H_2_O_2_ and have higher ρI_sp_ values than UDMH, they were not confirmed by drop-test experiments. Among all these ILs, IL-17 exhibited the shortest IDT of 18.5 ms and a competitive I_sp_ of 258 s, but it was in a solid state at ambient temperatures. IL-18 to IL-21 had high densities and self-ignition behavior owing to the solvents (DBN and DBU), which were the main components of the fuels. These limitations highlight two research priorities for borohydride-rich anion architectures: experimentally validating computationally predicted hypergolic candidates, and developing novel ionic liquids that combine room-temperature liquidity, solvent-free formulations, and spontaneous ignition capabilities.

### 2.2. ILs Based on Thiocyanate Anions

Recently, researchers have discovered that ionic liquids based on thiocyanate anions can also self-ignite with H_2_O_2_. In 2021, Lauck et al. conducted drop tests with two different ionic liquids based on thiocyanate anions [1-ethyl-3-methylimidazolium thiocyanate ([EMIM][SCN], IL-22) and 1-butyl-3-methylimidazolium thiocyanate ([BMIM][SCN], IL-23)] and H_2_O_2_ [21]. The structures of IL-22 and IL-23 ionic liquids are shown in Figure 4. Both ionic liquids can be ignited with 96.1% H_2_O_2_, exhibiting average IDTs of 31.7 ms for IL-22 and 45 ms for IL-23. Additionally, the theoretical I_sp_ calculated with NASA CEA for the two ionic liquids was approximately 318 s. Owing to their higher fuel density, the ρI_sp_ was enhanced by 10% compared to the conventional hypergolic propellant combination of monomethyl hydrazine/dinitrogen tetroxide (MMH/NTO). These results demonstrate that ionic liquids based on thiocyanate anions combined with high-concentration H_2_O_2_ hold great potential as viable alternatives to existing hypergolic propellant systems.

In 2021, Ricker et al. proposed six pyridinium- and pyrrolidinium-based thiocyanate ionic liquids (IL-24 to IL-29) as new fuel candidates with 97.4% H_2_O_2_ as an oxidizer, as shown in Figure 4 [22]. The results indicated that all ionic liquids with pyridinium-based cations showed shorter IDTs than those with cationic pyrrolidinium-based frameworks. Within the groups of the same heterocycles, shorter alkyl chains in the cation positively influenced the ignition behavior of the hypergolic propellants, while the degree of unsaturation in the side chains had no significant effect on IDTs. Furthermore, all studied fuels exhibited higher theoretical ρI_sp_ compared to the conventional hypergolic propellant combination of MMH/NTO. IL-24 demonstrated the shortest IDT (26.8 ms), the lowest viscosity (27.5 mPa s), a high density (1.13 g cm^−3^), and excellent thermal stability (T_d_ of 247 °C). This study not only identifies IL-24 as a promising candidate for future orbital propulsion systems but also provides new insights into the role of cationic structures in ionic liquids for hypergolic propellant combinations with H_2_O_2_.

The following year, Ricker et al. synthesized seven protic ionic liquids (IL-30 to IL-36) containing thiocyanate anions, as illustrated in Figure 5 [23]. Theoretical calculations using the NASA CEA code revealed that all ionic liquids with 97% H_2_O_2_ exhibited higher ρI_sp_ compared to the conventional toxic propellant MMH/NTO. The results showed that the IDTs increased with increasing chain length at the same position, which can be attributed to steric effects. Among these, IL-30 demonstrated an exceptionally short average IDT of 7.3 ms. However, since IL-30 is solid at ambient conditions, a blend of 35 wt% IL-30 and 65 wt% IL-22 (liquid at room temperature) was developed, forming a novel liquid bipropellant named HIM_35. This mixture achieved a low IDT of 16.7 ms. Notably, these fuels avoid the use of catalytic transition metals, boron-based compounds (which generate solid combustion residues), or air/moisture-sensitive hydrides. Consequently, they represent highly promising alternatives for environmentally friendly hypergolic bipropellants in space propulsion systems.

In 2024, Stölzle et al. synthesized three trialkylsulfonium thiocyanate ionic liquids (IL-37 to IL-39, Figure 6) and evaluated their potential as hypergolic fuels with high-concentration H_2_O_2_ [24]. Theoretical combustion calculations revealed that IL-39 achieved the highest I_sp_ of 314 s at O/F of 4.5, while IL-37 exhibited superior ρI_sp_ of 429 s g cm^−3^ at O/F = 3.9, attributed to its exceptional density. Hypergolic ignition behavior was further investigated through drop tests under ambient conditions and at reduced fuel temperatures (1 °C and −25 °C). All three propellant combinations maintained reliable ignition even at −25 °C, though lower temperatures prolonged IDTs. Notably, IL-38 demonstrated the shortest IDT, with average values of 30.8 ms at ambient conditions, increasing marginally to 32.8 ms at 1 °C and significantly to 51.8 ms at −25 °C, highlighting temperature-dependent kinetic limitations in thiocyanate-based systems.

The physicochemical properties of all ionic liquids based on thiocyanate anions are shown in Table 3. These ionic liquids exhibit rapid hypergolic ignition with high-concentration H_2_O_2_, achieving a minimum IDT of 7.3 ms and a high I_sp_ of up to 320 s. Cationic structural design plays a critical role in their performance. Shorter alkyl chains on pyridinium, pyrrolidinium, and imidazolium cations significantly reduce IDTs. However, these systems are heavily dependent on the high-concentration H_2_O_2_, and lower temperatures markedly degrade the ignition efficiency (e.g., IL-37’s IDT increases to 83.6 ms at −25 °C). Notably, trialkylsulfonium-based IL-38 defies the alkyl chain length trend, achieving the shortest IDT of 30.8 ms despite longer substituents, suggesting potential asymmetry effects in cationic frameworks.

Future investigations should prioritize these objectives: the first is elucidating the structure–ignition relationship in trialkylsulfonium cations to clarify the role of asymmetry, and the second is expanding the library of hypergolic thiocyanate-based ionic liquids through systematic cation–anion pairing strategies. Additionally, current systems require high-concentration H_2_O_2_ for spontaneous ignition, so further research is essential to evaluate their compatibility with low-concentration H_2_O_2_, which could enhance practicality and environmental adaptability.

## 3. Ionic Liquid–Hydrogen Peroxide Liquid Propellant Requiring Promoters

### 3.1. Promoters for ILs Based on Borohydride-Rich Anions and Hydrogen Peroxide

In 2018, Chinnam et al. developed novel iodine-rich hypergolic promoters (ILP-1 and ILP-2) to enable rapid ignition between IL-3 and H_2_O_2_ [25]. As illustrated in Figure 7, these promoters were synthesized to address the extremely long IDT (>30 s) observed for IL-3 with 70% H_2_O_2_ in the absence of a promoter. Remarkably, adding 8 wt% ILP-2 reduced the IDT to 45 ms with 70% H_2_O_2_ and further to 17 ms with 95% H_2_O_2_. In order to better understand the promoter effect of the [B_12_I_12_]^2−^ anion on this oxidation process, theoretical calculations were conducted, and they revealed that the [B_12_I_12_]^2−^ anion in ILP-2 significantly lowered the Gibbs free energies of intermediates during the initial oxidation steps, thereby accelerating substrate decomposition. Thus, organometallic salts containing ferrocene ([FcCH_2_NEtMe_2_]^+^) or copper ([Cu(en)_2_(CH_3_CN)_2_]^2+^) exhibited enhanced catalytic activity when paired with [B_12_I_12_]^2−^, forming unique bifunctional promoters. This multifunctional design demonstrates a breakthrough in utilizing safer 70% H_2_O_2_ (instead of highly concentrated H_2_O_2_) for hypergolic systems, offering a sustainable pathway for developing green oxidizers in space propulsion.

In the same year, Wang et al. synthesized and characterized four iodocuprate-based ionic liquid promoters (ILP-3 to ILP-6) to achieve rapid ignition between fuels (IL-3 or [MIM][BH_3_] (IL-40)) and 95% H_2_O_2_ [26]. The synthesis routes of these promoters are shown in Figure 8. Among them, ILP-5 emerged as the most promising candidate. Firstly, its decomposition temperature exceeded that of the fuels, ensuring stability in fuel–promoter mixtures for weeks. Secondly, it maintained the mixture’s viscosity at 50 mPa s and remained homogeneously dispersed in IL-3 for over four weeks without degradation of either the promoter or fuel. Furthermore, with 10 wt% ILP-5, the IDTs of IL-3 and IL-40 were shortened to 24 ms and 14 ms, respectively. Regarding the reaction mechanism of the catalytic process, the promoters can react with H_2_O_2_, accelerating substrate decomposition. In summary, the exceptional stability and catalytic efficiency of the newly synthesized promoters, especially of ILP-5, highlight their potential to advance the development of green bipropellant systems using H_2_O_2_ for space propulsion applications.

In 2019, Wang et al. synthesized a series of hypergolic ionic liquids (IL-41 to IL-49) containing the cyano (1H-1,2,3-triazole-1-yl) dihydroborate anion [27]. The synthesis routes of these ionic liquids are shown in Figure 9. While the pure ionic liquids did not spontaneously ignite with 90% H_2_O_2_, the addition of 15 wt% iodine (I_2_, ILP-7) enabled the hypergolic ignition of IL-42 to IL-43. This result demonstrated that iodine acted as an efficient catalyst to promote reactions between boron-containing ionic liquids and H_2_O_2_. Such catalytic capability positions iodine as a promising candidate for green propellant systems based on ionic liquid–H_2_O_2_ combinations.

In 2020, Bhosale et al. investigated the use of various promoters (ILP-8 to ILP-12) to enhance the hypergolic ignition of ionic liquids (IL-3 and IL-4) with 95% H_2_O_2_ [5]. The structures of ILP-8 to ILP-11 are shown in Figure 10. At a concentration of 5 wt%, these promoters reduced the IDTs of IL-3 to 139 ms (ILP-8), 395 ms (ILP-9), 887 ms (ILP-10), and 73 ms (ILP-11). 1,3-dimethyl imidazolium copper iodide ([diMIM]_n_[Cu_2_I_3_]_n_, ILP-12) was a newly synthesized promoter, shown in Figure 11, and its physicochemical properties were further enhanced. Adding 5 wt% ILP-12 shortened the IDT of IL-3 to 87 ms. Notably, increasing the ILP-12 concentration from 2 to 15 wt% in IL-3 significantly reduced the IDT of IL-3 from 126 to 29 ms and 47 to 13 ms under oxidizer-rich and fuel-rich conditions, respectively. Meanwhile, higher ILP-12 concentrations effectively improved the density and viscosity of the fuel mixture. These findings demonstrate that additive-promoted hypergolic combustion offers a viable pathway to replace conventional toxic propellant systems.

In 2020, addressing the poor solubility of existing hypergolic promoters, Wang et al. synthesized four novel promoters (ILP-13 to ILP-16) with ultrahigh solubility in energetic ionic liquids based on the principle of similarity compatibility [28]. Their structures are shown in Figure 12. To systematically compare the roles of anions and cations, two reference promoters were designed: The first is the “cation active” copper-free reference promoter ILP-15, whereas the second is an “anion active” iron-free reference promoter ILP-16. The ignition effect of these promoters on ionic liquid fuels IL-3 and IL-5 was studied. Adding 10 wt% ILP-13 or ILP-14 reduced the IDT of IL-3 to 38 ms and 31 ms, respectively, while for IL-5, the IDTs were 89 ms and 56 ms under the same conditions. In addition, ILP-13 and ILP-14 exhibited excellent solubility and chemical stability in ionic liquids. This study highlights that incorporating metals (e.g., Fe, Cu) into the anion of the promoter plays a predominant role in reducing IDTs, providing critical insights for designing next-generation hypergolic systems.

In the same year, Bhosale et al. conducted research on the use of sodium iodide (NaI, ILP-17) as a promoter to enhance the ignition performance of the hypergolic fuels IL-3 and IL-40, with 95% H_2_O_2_ serving as the green oxidizer [29]. They prepared seventeen fuel blends by mixing ILP-17 with the fuels (Table 4), where HF-1 (pure IL-3) and HF-9 (pure IL-40) exhibited IDTs of >1000 ms and 392.5 ms, respectively. The optimal blend, HF-10, consisting of 3 wt.% ILP-17 and 97 wt.% IL-40, achieved a significantly shorter IDT of 25.3 ms while maintaining an I_sp_ of 240 s. Additionally, furfuryl alcohol was introduced to improve the solubility of ILP-17 in IL-3 and IL-40, but no notable reduction in IDT was observed. This study highlights the potential of sodium iodide as a promoter in developing green hypergolic bipropellant systems, offering a promising direction for environmentally friendly and high-performance propulsion technologies.

In 2021, Bhosale et al. developed two new hypergolic energetic copper (II) promoters, [Cu^II^(1-H-imidazole)_4_(BH_3_CN)][BH_3_CN] (ILP-18) and [Cu^II^(1-methyl imidazole)_4_(BH_3_CN)_2_] (ILP-19), for green propellant systems, with their synthetic pathway illustrated in Figure 13 [30]. These transition metal complexes demonstrated remarkable hypergolic reactivity with 95% H_2_O_2_, achieving IDTs of 3.75 ms and 8.50 ms for ILP-18 and ILP-19, respectively. To evaluate their practical application potential, ILP-18 was dissolved at 13 wt% in three distinct fuel systems: ionic liquid IL-3, tetraglyme (TG) solvent, and an equal-weight mixture of IL-3 and TG. Drop contact tests with 95% H_2_O_2_ revealed IDTs of 9.5 ms, 9.0 ms, and 7.8 ms for these systems, respectively, indicating a significant reduction compared to the pure ionic liquids and highlighting the superior performance of the IL-3/TG hybrid system. These findings underscore the promising potential of hypergolic ionic liquid fuel blends incorporating optimized amounts of energetic promoters and organic solvents as a viable alternative for green bipropellant fuels.

In 2024, to meet the performance requirements for rocket engine fire testing, Bhosale et al. investigated a novel hypergolic fuel blend designated as ILethCu01, composed of 9 wt% ILP-18 promoter in an equal-weight mixture of IL-3 (primary fuel) and ethanol (co-fuel) [31]. This ternary system demonstrated exceptional compatibility with 95 wt% H_2_O_2_, exhibiting a remarkably low viscosity of 3.85 mPa s and a favorable density of 0.90 g cm^−3^. Theoretical performance calculations revealed an I_sp_ of 317 s and a ρI_sp_ of 403 s g cm^−3^, surpassing conventional hypergolic systems. Crucially, drop-test experiments recorded an average IDT of 7.50 ms upon contact with 95 wt% H_2_O_2_, confirming rapid hypergolic initiation. These integrated properties—combining fluidity, energy density, and instantaneous ignition—establish ILethCu01/H_2_O_2_ as a leading candidate for next-generation green hypergolic propulsion systems.

In the same year, Seo et al. investigated the development of low-toxicity hypergolic propellants by enhancing triglyme-based fuels with ILP-18 and IL-3 [32]. Their study evaluated the IDTs of triglyme and triglyme–ionic liquid blends (TriGILs) containing varying ILP-18 concentrations, tested with 70, 90, and 95% H_2_O_2_. The 13Cu-TriGIL (TriGIL with 13% ILP-18) demonstrated exceptional performance, achieving an 8.0 ms IDT with 95% H_2_O_2_ compared to 18.7 ms for triglyme alone. The ionic liquid integration also improved cryogenic stability, reducing the blend’s freezing point from −45 °C (pure triglyme) to below −80 °C. The optimized fuel exhibited favorable propulsion characteristics, including a density of 1.017 g cm^−3^ and viscosity of 26.42 mPa s. Theoretical calculations revealed that 13Cu-TriGIL with 95% H_2_O_2_ exhibited a 9.7% higher ρI_sp_ than MMH/NTO, despite a 2.1% lower I_sp_. This work underscores triglyme–ionic liquid blends as promising eco-friendly alternatives to toxic hypergolic propellants, combining rapid ignition, enhanced thermal stability, and reduced hazards while maintaining competitive performance metrics.

In 2021, Zhao et al. designed, prepared, and comprehensively characterized six halogen-free energetic complexes (ILP-20 to ILP-22 and ILP-26 to ILP-28) with their synthesis scheme illustrated in Figure 14 [33]. These compounds exhibited high densities (1.180 to 1.373 g cm^−3^) and robust thermal stability, displaying decomposition temperatures which ranged from 165.1 to 269.8 °C. Their catalytic performance was evaluated for the hypergolic reaction between ionic liquid IL-3 and 90% H_2_O_2_. Pure IL-3 exhibited an initial IDT exceeding 4000 ms with 90% H_2_O_2_, while the addition of 10 wt% promoter loading drastically reduced this parameter. Notably, ILP-20 and ILP-26 demonstrated superior catalytic performance, achieving IDTs of 37 ms and 31 ms, respectively. In 2022, as a continuation, Zhao et al. designed and comprehensively characterized six other halogen-free energetic complexes (ILP-23 to ILP-25 and ILP-29 to ILP-31) [34]. Their synthesis scheme is shown in Figure 14. Their densities ranged from 1.157 to 1.347 g cm^−3^. Their thermal decomposition temperatures ranged from 169.6 to 255.9 °C. After adding 10 wt.% of prepared promoters to IL-3, the shortest IDT achieved was 94 ms.

These twelve complexes comprise two distinct categories: six borohydride-rich isomeric energetic complexes (ILP-20 to ILP-25) and six higher-density boron-free energetic complexes (ILP-26 to ILP-31). These materials share structural similarities while exhibiting differentiated ignition properties, providing an ideal platform for elucidating the catalytic ignition mechanisms between ionic liquids and high-concentration H_2_O_2_. The ignition process occurs through two sequential phases: H_2_O_2_ and the catalyst come into contact, and copper ions rapidly catalyze the decomposition of H_2_O_2_, generating highly reactive free radicals accompanied by significant exothermic release; this thermal energy then elevates the system temperature while the radicals initiate ionic liquid combustion, synergistically accelerating reaction kinetics to achieve spontaneous ignition. Notably, these halogen-free complexes demonstrate exceptional practical advantages through simple synthesis methods, high yields and superior performance metrics, positioning them as promising hypergolic promoters for green bipropellant systems that simultaneously address energy density requirements and environmental sustainability concerns.

Few borohydride-rich ionic liquids exhibit spontaneous ignition with H_2_O_2_, necessitating the use of diverse promoters to enhance their hypergolic performance, as summarized in Table 5. Analysis reveals that these promoters primarily consist of metal elements (such as copper and iron), borides, and iodides, which reduce the Gibbs free energy barrier or catalyze substrate decomposition, thereby achieving lower IDTs. Notably, ILP-18 at 13 wt% concentration paired with IL-3 and 95% H_2_O_2_ delivered the highest performance, achieving the shortest IDT of 9.5 ms. ILethCu01 (9% ILP-18) achieves an ultralow IDT of 7.5 ms with a viscosity of 3.85 mPa·s. Promoter solubility in ionic liquids is critical, prompting strategies such as blending organic solvents into ionic liquid fuels to improve solubility. Experimental results confirm that solvent-optimized ionic liquid mixtures effectively reduce IDTs due to enhanced dissolution of active components. For instance, replacing pure IL-3 with an equal-weight blend of IL-3 and TG (an organic solvent) shortened the IDT from 9.5 ms to 7.8 ms, demonstrating the synergistic benefits of solubility-driven formulation design. Meanwhile, increased viscosity may impair combustion efficiency. The viscosity of IL-5 with ILP-14 rises from 28 to 47 mPa s. Future efforts should prioritize designing multifunctional promoters that balance catalytic activity with physicochemical properties, alongside eco-friendly and cost-effective synthesis routes.

### 3.2. Promoters for ILs Based on Thiocyanate Anions and Hydrogen Peroxide

The previous text mentioned that Lauck et al. conducted drop tests with two different ionic liquids based on thiocyanate anions (IL-22 and IL-23) and 96.1% H_2_O_2_ in 2021 [21], reporting average IDTs of 31.7 ms and 45 ms, respectively. To optimize performance, copper thiocyanate (ILP-32) was introduced as a catalytic additive into IL-22. Promoted fuels containing different amounts of ILP-32 were systematically evaluated, revealing a non-linear relationship between additive concentration and ignition efficiency: the minimum delay of 13.9 ms was achieved at 5 wt% ILP-32, beyond which delays increased marginally. However, this enhancement came with trade-offs—fuel viscosity rose from 20 mPa s (pure IL-22) to 29.6 mPa s at 5 wt% ILP-32, while metallic additives introduced secondary challenges including I_sp_ degradation and combustion chamber fouling risks from particulate formation. These results highlight the critical balance required between ignition kinetics and operational practicality in catalytic hypergolic fuel design.

In the same year, Stützer et al. expanded the investigation of ILP-32’s catalytic effects on the hypergolic reaction between IL-22 and 96.1% H_2_O_2_ by conducting laboratory-scale drop tests and flame emission spectroscopy to analyze combustion chemistry [35]. Their spectral analysis revealed that dissolving 5 wt% ILP-32 in IL-22 not only reduced the IDT (from 31 ms for pure IL-22 to 15 ms) but also elevated flame temperatures during rapid combustion. This dual functionality—enhancing both ignition kinetics and combustion intensity—demonstrates the critical role of ILP-32 in optimizing hypergolic performance. The study provides mechanistic insights into how transition metal additives modulate redox dynamics at the fuel–oxidizer interface, advancing the rational design of catalytic promoters for high-efficiency green propellants.

In 2022, Lauck et al. investigated the performance of a novel green hypergolic propellant, HIP_11, composed of 97% H_2_O_2_ as the oxidizer and a fuel blend of IL-22 with 5% ILP-32, aiming to evaluate the ignition reliability, combustion efficiency, and stability in a small “battleship” thruster designed for robustness and repeated testing [36]. Initial injector tests demonstrated that increasing the ILP-32 concentration from 1% to 5% markedly reduced IDTs to 13.9 ms, ensuring consistent hypergolic initiation. Subsequent hot-fire campaigns achieved stable combustion at chamber pressures exceeding 7 bar, delivering 93–98% combustion efficiency with exceptionally low pressure oscillations (1.1–1.7% of chamber pressure) and rapid rise times under 13 ms. This study marked the first successful demonstration of an ionic liquid–hydrogen peroxide system in operational thruster conditions, showing HIP_11’s viability as a low-toxicity alternative to conventional propellants. The system’s simplicity, repeatable ignition performance, and efficiency comparable to established hypergolic technologies highlight its potential for practical aerospace applications while addressing environmental and safety concerns.

Li et al. investigated the hypergolic ignition behavior of imidazolium thiocyanate-based ionic liquids blended with ethylene glycol (ILP-33) and propylene glycol (ILP-34) as additives in 90% H_2_O_2_ [37]. Their study revealed a counterintuitive trend: increasing the molar ratio of either glycol additive led to longer IDTs compared to the pure ionic liquid baseline. At fixed collision velocities, higher additive loadings further degraded combustion efficiency, with IDTs scaling inversely with glycol concentration. Notably, blends containing elevated ILP-33 or ILP-34 exhibited weakened flame propagation and reduced energy release rates, conclusively demonstrating that glycol additives impair, rather than enhance, the combustion performance of thiocyanate ionic liquids in concentrated H_2_O_2_ systems. This critical finding challenges conventional assumptions about oxygenated co-solvents, underscoring the need for alternative promoter strategies in hypergolic fuel design.

Park et al. investigated the impact of oxidizing additives, such as LiNO_3_ (ILP-35) and NH_4_NO_3_ (ILP-36) [38], on the physical properties and ignition performance of green hypergolic systems composed of ionic liquid fuels (IL-22 and IL-23) and H_2_O_2_. While the additives negatively influenced the theoretical performance metrics, specifically reducing I_sp_ and ρI_sp_, they significantly improved the low-temperature stability of the oxidizer mixtures. Notably, the addition of just 5 wt% ILP-35 to 95% H_2_O_2_ lowered the freezing point to −30 °C, demonstrating its exceptional efficacy. Furthermore, the oxidizing additives markedly enhanced ignition performance, with ILP-35 outperforming ILP-36 in drop tests. This improvement was amplified at lower H_2_O_2_ concentrations: although ILP-36 degraded ignition performance when combined with 90–95 wt% H_2_O_2_, the hypergolicity limit was extended when ILP-35 was incorporated into 60% H_2_O_2_ and paired with IL-23. These findings highlight the potential of nitrate salts as functional additives for tailoring H_2_O_2_-based oxidizers to achieve balanced cryogenic stability and ignition characteristics, despite their trade-offs in theoretical propulsion performance.

The physicochemical properties of thiocyanate-rich ionic liquid–H_2_O_2_ propellant systems with promoters are detailed in Table 6, revealing limited promoter diversity for thiocyanate-based formulations. Among the tested promoters (ILP-32 to ILP-36), inorganic salts (ILP-32, ILP-35 and ILP-36) effectively enhanced hypergolic ignition between thiocyanate-rich ionic liquids and high-concentration H_2_O_2_, whereas organic promoters (ILP-33 and ILP-34) exhibited that the addition of them correlated with prolonged IDT, disqualifying them for this propellant system. Notably, ILP-32 at a 5% concentration achieved the shortest IDT of 13.9 ms, outperforming the other two nitrates ILP-35 and ILP-36. Further analysis suggests that ILP-35 exhibits superior catalytic performance compared to ILP-36, likely due to its specific metal ion content, which accelerates reaction kinetics. These findings underscore the importance of continued exploration of inorganic salt promoters to optimize hypergolic performance in bisulfate ionic liquid–hydrogen peroxide systems, highlighting inorganic additives as a priority for future research.

### 3.3. Promoters for ILs Based on Dicyanamide Anions and Hydrogen Peroxide

In addition to borohydride-rich and thiocyanate-rich ionic liquids, dicyanamide anion-based ionic liquids are widely employed as propellant fuels. In 2014, Schneider et al. pioneered the use of metal-containing ionic liquids as hypergolic bipropellants to enhance ignition performance [39]. Their work focused on synthesizing 1-butyl-3-methylimidazolium chloroferrate ([BMIM][FeCl_4_], ILP-37, Figure 15), which acted as a hypergolic promoter in reactions between ionic liquids and 98% H_2_O_2_. When tested with diaminomethylene azide dicyanamide ([DMAZ][N(CN)_2_], IL-50, Figure 15) and H_2_O_2_, ILP-37 at a concentration of 8 wt% demonstrated its effectiveness as a hypergolic catalyst by significantly reducing IDTs to 110 ms with 90% H_2_O_2_ and 130 ms with 98% H_2_O_2_. This improvement stems from the ability of metal-containing ionic liquids to accelerate H_2_O_2_ decomposition, thereby enabling rapid ignition in bipropellant systems. These findings established a novel strategy for designing high-performance hypergolic formulations by integrating catalytic metal ions into ionic liquid frameworks, offering a pathway to optimize both reaction kinetics and system reliability.

In 2017, Weiser et al. conducted a study investigating the ignition performance of 1-allyl-3-methylimidazolium dicyanamide ([AMIM][N(CN)_2_], IL-51, Figure 15) when combined with high-concentration H_2_O_2_ and a copper-based catalyst at a concentration of 15 wt% [40]. The team evaluated their ignition behavior through ignition tests using 95% H_2_O_2_, 90% H_2_O_2_, 80% H_2_O_2_, and 70% H_2_O_2_, which demonstrated remarkably short IDTs of 9 ms, 11 ms, 22 ms, and 66 ms, respectively. Recognizing the practical limitations of pure high-concentration H_2_O_2_, including long-term stability issues and low-temperature solidification, the researchers investigated additive-modified oxidizer formulations. They incorporated ammonium nitrate (AN, ILP-38), ammonium dinitramide (ADN, ILP-39), and urea peroxide (Urea, ILP-40) to improve thermal stability, lower melting points, and optimize oxygen balance and combustion efficiency. This investigation led to the development of 40 distinct oxidizer mixtures with varying mass ratios of H_2_O_2_, ILP-38, ILP-39 and ILP-40, all paired with IL-40 as the fuel component. The resulting IDTs across all formulations showed significant variation, spanning from 15 ms to nearly 200 ms, highlighting the critical role of oxidizer composition in tuning hypergolic performance. These findings underscore the exceptional reactivity of ionic liquid–oxidizer combinations when using high-purity H_2_O_2_, while simultaneously providing a framework for tailoring ignition characteristics through strategic additive selection and formulation optimization.

The previous text mentioned that Wang et al. synthesized promoters ILP-13 and ILP-14 in 2020, in order to investigate their ignition-enhancing effects on the ionic liquid fuels IL-3 and IL-10, while also evaluating their performance with 1-ethyl-3-methylimidazolium dicyanamide ([EMIM][N(CN)_2_], IL-52) [28]. Their experiments revealed that blending IL-52 with 10% ILP-14 reduced the IDT to 42 ms when combined with 95% H_2_O_2_, whereas the same concentration of ILP-13 resulted in a slightly longer delay of 54 ms. However, the incorporation of these promoters introduced trade-offs: fuel mixtures exhibited marginally increased density and viscosity, with IL-52’s inherently high base viscosity (69 mPa s) further compromising overall performance. Consequently, the modified IL-52 formulations underperformed compared to IL-3 and IL-5 in comprehensive evaluations, demonstrating that promoter-enhanced ignition efficiency must be balanced against detrimental physical property changes in hypergolic fuel design.

In 2022, Wang et al. provided a strategy for designing promoters by the synergy of cations and anions to seek green bipropellants [41]. They designed two difunctional promoters containing imidazolium-substituted borohydride cations and iodocuprate anions for hypergolic ignition of ionic liquids without BH_4_^−^/BH_3_CN^−^ anions and 90% H_2_O_2_, which are [Bis(1-ethyl-1H-imidazole-3-ium-3-yl)dihydroboronium][Cu_3_I_4_] (ILP-41) and [Bis(1-allyl-1H-imidazole-3-ium-3-yl) dihydroboronium]_2_[Cu_4_I_6_] (ILP-42). The synthesis scheme of ILP-41 and ILP-42 is shown in Figure 16. The study employed the ionic liquids [AMIM][N(CN)_2_] (IL-40) and [BMIM][N(CN)_2_] (IL-53) as fuels. Testing revealed that ILP-41 and ILP-42 had strong ignition-promoting potential. IL-40 with 5 wt% ILP-41 achieved an IDT of 75.0 ms, while increasing ILP-42 concentrations from 3 to 20 wt% progressively reduced IDTs, culminating in a 34.0 ms delay with 20 wt% ILP-42. For IL-53, 5 wt% additions of ILP-41 and ILP-42 yielded delays of 161 ms and 112.5 ms, respectively. Their efficacy stems from their dual-action mechanism: iodocuprate anions catalyze the exothermic decomposition of H_2_O_2_, while borohydride-functionalized cations act as ignition initiators. This work demonstrates how strategic cation–anion coordination in promoter design can advance environmentally sustainable hypergolic propellant systems.

In 2024, Liao et al. introduced an innovative strategy to enhance hypergolic ignition in ionic liquid fuels using H_2_O_2_ as the oxidizer and copper-based ionic liquid promoters (ILP-43 to ILP-47) structurally tailored to mimic the cation composition of the fuel molecules (IL-40, IL-52 to IL-55), as illustrated in Figure 17 [42]. Leveraging the “like dissolves like” principle, these fuel–promoter pairs achieved full miscibility across most ratios, though ILP-47 exhibited slightly reduced solubility, requiring a 12% concentration for complete mixing with IL-52. Testing demonstrated a clear correlation between solubility and ignition performance: IL-52 paired with ILP-47 showed a prolonged delay of 59 ms, while combinations with the other four promoters all achieved sub-40 ms ignition times. ILP-43 emerged as the most effective promoter, enabling IL-52 to reach a remarkably short 16 ms delay. This superiority extended to other fuels, with all four ionic liquids paired with ILP-43 consistently igniting in under 40 ms when combined with 90% H_2_O_2_. The study underscores how strategic alignment of promoter–fuel molecular structures optimizes miscibility and ignition efficiency, advancing the development of high-performance hypergolic bipropellant systems.

In summary, the physicochemical properties of dicyanamide-rich ionic liquid–hydrogen peroxide propellant systems with promoters are shown in Table 7. Promoters for dicyanamide-rich ionic liquids paired with H_2_O_2_ have evolved to incorporate catalytic metal ions (such as iron and copper) and reactive functional groups (such as borohydride and iodocuprate) within their ionic frameworks. These promoters, such as ILP-37, ILP-41, and ILP-42, enhance ignition by dual mechanisms: metal anions catalyze the exothermic decomposition of H_2_O_2_, while functionalized cations initiate rapid fuel oxidation. Structural alignment between promoters (ILP-43 to ILP-47) and fuels (IL-51 to IL-55) further optimizes miscibility and ignition efficiency, achieving IDTs below 40 ms. Despite these advancements, inherent trade-offs persist between catalytic activity and physicochemical properties, necessitating continued innovation. Future efforts should focus on designing eco-friendly, multifunctional promoters with tailored cation–anion synergies, balanced physicochemical properties, and scalable synthesis routes to advance sustainable hypergolic bipropellant systems.

### 3.4. Promoters for ILs Based on Other Anions and Hydrogen Peroxide

The previous text mentioned that Schneider et al. synthesized the promoter [BMIM][FeCl_4_] (ILP-37), which is used in the self-ignition reaction between ionic liquids and 98% H_2_O_2_ in 2014 [39]. In fact, in addition to the ionic liquids [DMAZ][N(CN)_2_] (IL-50), ILP-37 also promoted hypergolic ignition between many other ionic liquids such as 1-butyl-3-methylimidazole azide ([BMIM][N_3_], IL-56), 2-hydroxyethylhydrazine nitrate ([HEH][NO_3_], IL-57) and dimethylethylene azide trifluoroacetic acid ammonium ([DMAZ][TFA], IL-58), and 98% H_2_O_2_. The IDTs were 170 ms, 50 ms, and 960 ms, respectively.

In 2019, addressing challenges posed by the high viscosity and poor miscibility of 1-butyl-3-methylimidazolium acetate ([BMIM][Ac], IL-59) with H_2_O_2_, Lauck et al. developed modified fuel blends by incorporating organic solvents [43]. Two formulations were tested: an 81.3% IL-59 mixture with 10% ethanol (ILP-48) and 8.7% MAT (ILP-49), and a 72.8% IL-59 composition with 19.4% ILP-48 and 7.8% ILP-49. Both mixtures significantly reduced viscosity and achieved full miscibility with H_2_O_2_, enabling reliable ignition in drop tests with consistent delay times averaging approximately 28 ms. However, this solvent-based approach partially compromised key advantages of ionic liquids—specifically their inherently low vapor pressures and high densities—highlighting the need to explore alternative ionic liquids that retain these beneficial properties while improving compatibility with oxidizers. The study underscores the delicate balance required between optimizing fuel–oxidizer interactions and preserving the intrinsic advantages of ionic liquid propellants in hypergolic system design.

To conclude, the physicochemical properties of other ionic liquid–hydrogen peroxide propellant systems with promoters are shown in Table 8. Notably, metal-ion promoters (such as ILP-37) have been extended to non-conventional ionic liquids (such as IL-56/57/58), but their performance exhibits significant variability. For instance, IDTs range from 50 ms for IL-57 to 960 ms for IL-58, underscoring the critical role of ionic liquid selection in system optimization. Solvent blending strategies, exemplified by combinations like IL-59+ILP-48/49, effectively reduce viscosity (from 97 to 37 mPa s), thereby improving fluidity.

## 4. Conclusions

In this paper, we review the recent trends and developments in green liquid propellants based on ionic liquid–hydrogen peroxide combinations and analyze in detail the synthesis and ignition performances of both self-igniting propellants and promoter-dependent propellants. The latest research has demonstrated that introducing functional promoters is a feasible strategy, which not only enables the rapid ignition of ionic liquids with high-concentration H_2_O_2_ but also significantly reduces IDTs and enhances the performance parameters of the ionic liquids.

The viscosity, I_sp_, melting point, stability, solubility, and cost of promoters are critical determinants of their practical applicability. First, viscosity directly influences the propellant fluidity, which governs injection efficiency and combustion performance. For example, adding ILP-32 to IL-22 elevates the viscosity from 20 to 29.6 mPa s, while blending IL-52 with ILP-13 or ILP-14 increases the viscosity from 69 to 78 and 84 mPa s, respectively. Elevated viscosity raises flow resistance, impairing injection efficiency, and poor mixing due to high viscosity can lead to incomplete combustion. To address this, solvent blending or the design of low-viscosity promoters (η < 50 mPa s) is essential. I_sp_ is an important index to measure the performance of propellants. Higher I_sp_ (>300 s) can enhance rocket engine efficiency, payload capacity, and velocity. Low melting points (T_m_ < room temperature) ensure propellant liquidity under cryogenic conditions, preventing solidification and ensuring reliable supply. Thermal stability (T_d_ > 200 °C) is vital for safe storage and consistent ignition in engines. The solubility of solid promoters is also a critical parameter in fuel formulation design. These promoters must be uniformly dispersed within the fuel to form a homogeneous system, ensuring sufficient contact with the fuel and enabling effective catalytic action. However, low solubility can lead to phase separation or sedimentation, resulting in uneven local concentrations. This imbalance may cause delayed or incomplete combustion, ultimately reducing specific impulse and compromising reliability. Cost considerations are equally critical: metal-containing promoters like ILP-41 offer superior catalytic efficiency but require complex synthesis and expensive metals, whereas metal-free alternatives like ILP-32 are cost-effective but less active. Future advancements must balance catalytic performance, economic viability, and environmental sustainability.

Among the promoters analyzed, ILP-18 emerges as the most promising candidate for rocket propulsion. As shown in Table 5, the IL-3/ethanol blend (ILethCu01) with 9% ILP-18 achieves an ultralow IDT of 7.5 ms, a low viscosity of 3.85 mPa s, and a high I_sp_ of 317 s. Notably, ILP-18 retains reliable hypergolic ignition even at extreme temperatures (the TriGIL system can operate at −80 °C). This catalyst functions through two complementary mechanisms: copper ions accelerate the decomposition of hydrogen peroxide, while borohydrides drive rapid redox reactions. These processes enable fast ignition and sustained combustion, meeting the practical requirements of rocket engines. These attributes position ILP-18 as the leading green propellant candidate for next-generation propulsion systems.

However, the exploration in this field is still insufficient, and more in-depth research should be carried out in the following aspects in the future. Firstly, the library of ionic liquids should be expanded by designing different cations and anions and identifying their structure–ignition relationships. Secondly, the library of promoters should be expanded and the connection between structure and catalytic performance explored. In conclusion, the development of new ionic liquid–hydrogen peroxide combinations will undoubtedly continue to advance the field of environmentally friendly chemical propulsion. In the future, we can expect a wide range of researchers to develop more novel high-performance ionic liquids and promoters, further driving progress in this area.

## Figures and Tables

**Figure 1 molecules-30-01789-f001:**
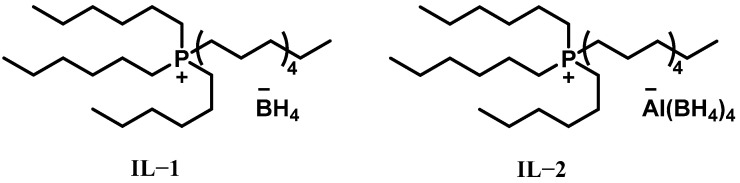
Structures of [THTDP][BH_4_] (IL-1) and [THTDP][Al(BH_4_)_4_] (IL-2) ionic liquids.

**Figure 2 molecules-30-01789-f002:**
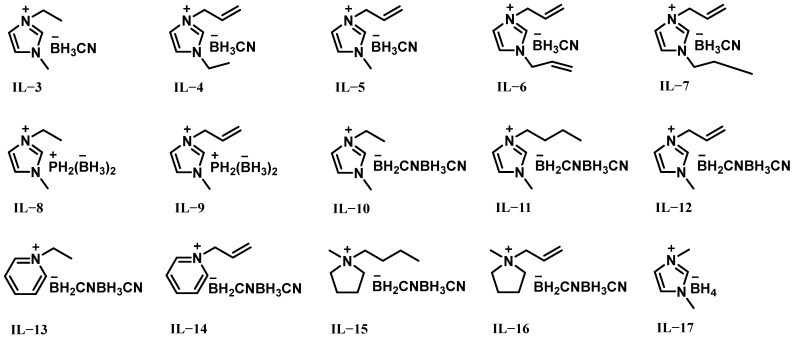
Structures of IL-3 to IL-17.

**Figure 3 molecules-30-01789-f003:**
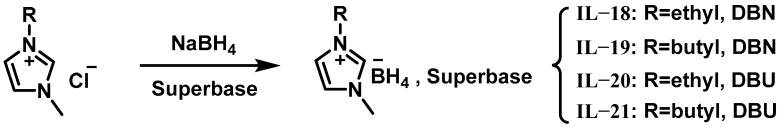
Synthesis method of the composite hypergolic fuels.

**Figure 4 molecules-30-01789-f004:**
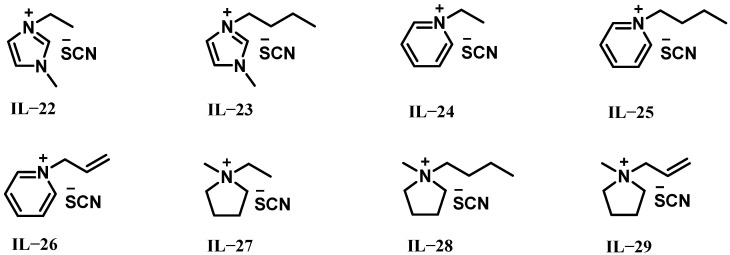
Structures of IL-22 to IL-29.

**Figure 5 molecules-30-01789-f005:**
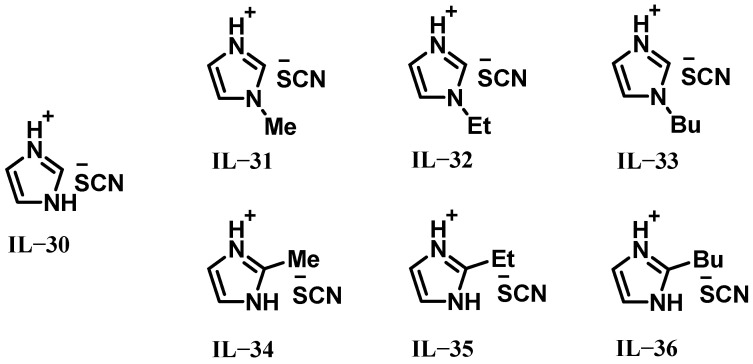
Structures of IL-30 to IL-36.

**Figure 6 molecules-30-01789-f006:**
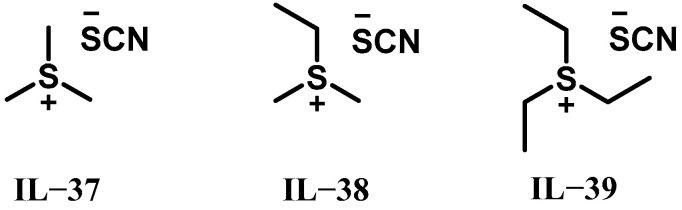
Structures of IL-37 to IL-39.

**Figure 7 molecules-30-01789-f007:**
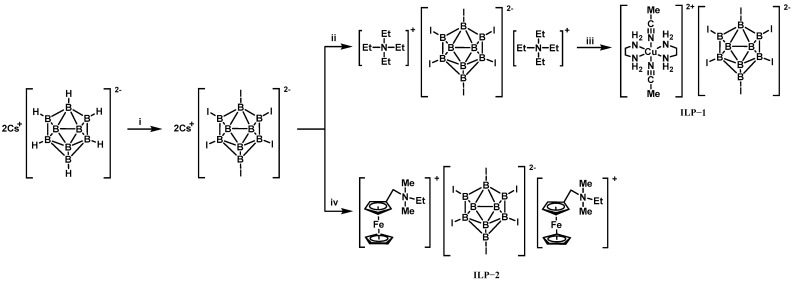
Synthesis scheme of ILP-1 and ILP-2. (Experimental conditions: (i) I_2_; (ii) tetraethylammonium bromide; (iii) [Cu(en)_2_(OH)_2_]; (iv) [FeCH_2_NEtMe][I]).

**Figure 8 molecules-30-01789-f008:**
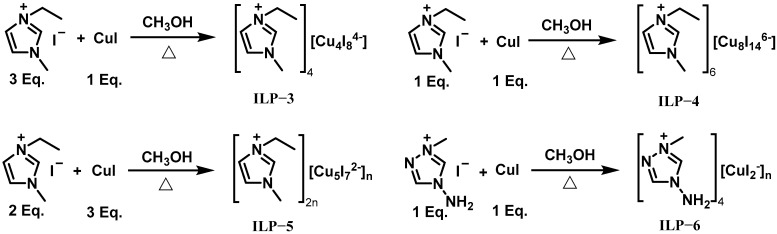
Synthesis routes of ILP-3 to ILP-6.

**Figure 9 molecules-30-01789-f009:**
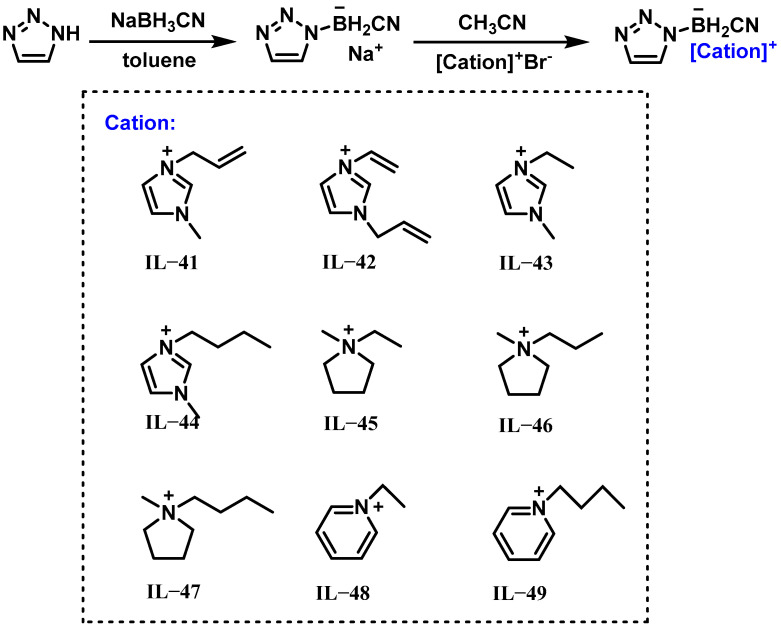
Synthesis of IL-41 to IL-49.

**Figure 10 molecules-30-01789-f010:**
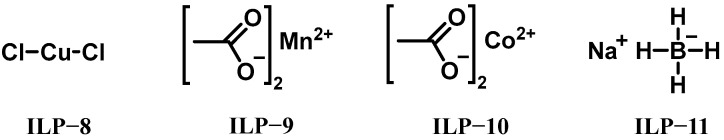
Structures of ILP-8 to ILP-11.

**Figure 11 molecules-30-01789-f011:**
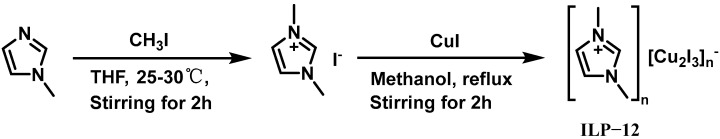
Synthesis of ILP-12.

**Figure 12 molecules-30-01789-f012:**
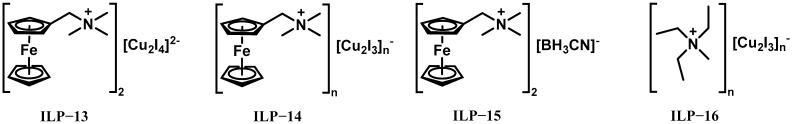
Structures of ILP-13 to ILP-16.

**Figure 13 molecules-30-01789-f013:**
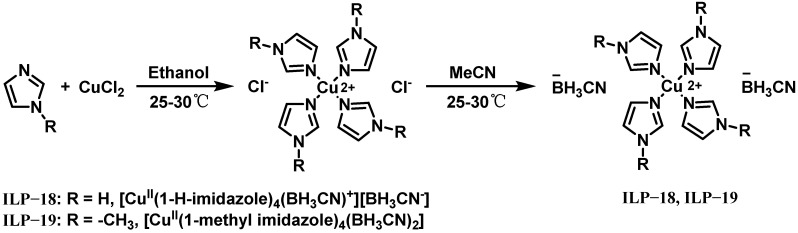
Synthesis of ILP-18 and ILP-19.

**Figure 14 molecules-30-01789-f014:**
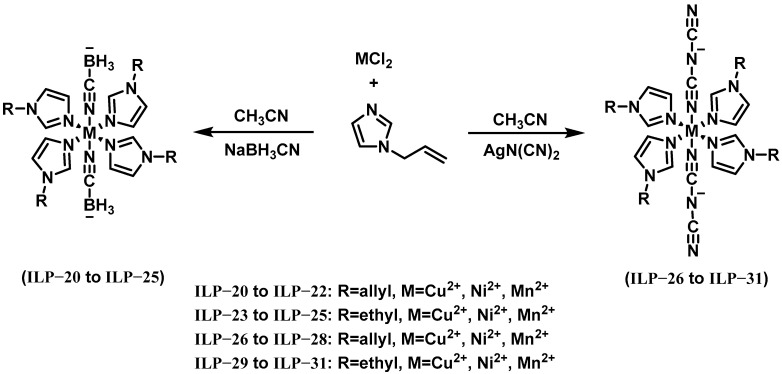
Synthesis of ILP-20 to ILP-31.

**Figure 15 molecules-30-01789-f015:**
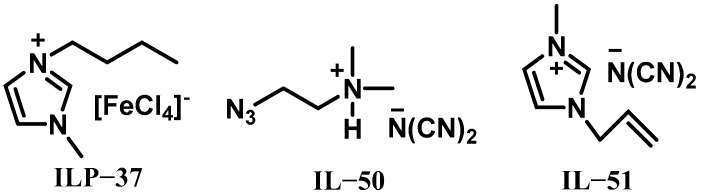
Structures of ILP-37 and IL-50.

**Figure 16 molecules-30-01789-f016:**
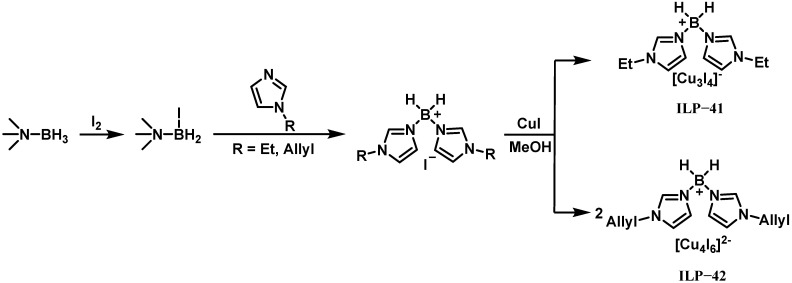
Synthesis of ILP-41 and ILP-42.

**Figure 17 molecules-30-01789-f017:**
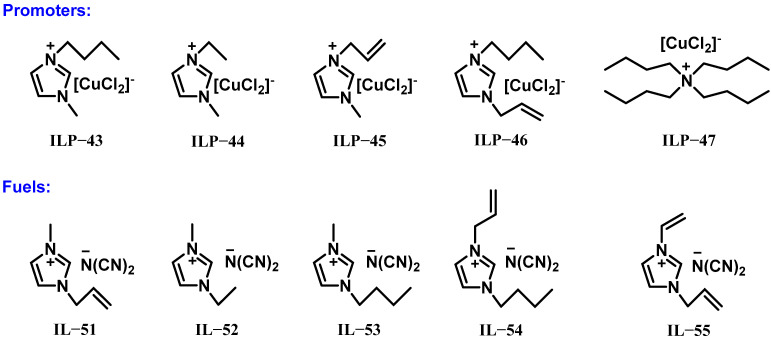
Structures of the promoters (ILP-43 to ILP-47) and the fuels (IL-51 to IL-55).

**Table 1 molecules-30-01789-t001:** Theoretical performances of 14 ionic liquids with H_2_O_2_.

Hypergolic Fuels	O/F ^a^	Vc ^b^ (m s^−1^)	I_sp_ ^c^ (s)	I_vac_ ^d^ (s)	ρ ^e^ (g cm^−3^)	ρI_sp_ ^f^ (s g cm^−3^)
IL-3	3.5	1670	244.1	273.3	1.310	319.8
IL-4	3.5	1663	243.1	272.1	1.299	315.9
IL-5	3.5	1673	244.5	273.7	1.303	318.6
IL-6	3.5	1668	243.9	272.9	1.301	317.4
IL-7	3.5	1669	244.0	272.9	1.294	315.8
IL-8	2.5	1664	243.2	271.9	1.234	300.2
IL-9	2.5	1666	243.5	272.2	1.240	301.9
IL-10	3.5	1646	240.6	269.6	1.288	309.9
IL-11	3.5	1648	240.9	269.7	1.297	312.5
IL-12	3.5	1650	241.1	270.2	1.292	311.4
IL-13	3.5	1646	240.6	269.4	1.291	310.7
IL-14	3.5	1642	240.0	268.8	1.294	310.7
IL-15	4.0	1659	242.4	271.5	1.286	311.8
IL-16	4.0	1661	242.7	272.0	1.292	313.7
UDMH	3.5	1694	247.7	276.9	1.224	303.3

^a^ optimum oxidizer-to-fuel ratio of propellant. ^b^ characteristic velocity of propellant. ^c^ specific impulse of propellant. ^d^ vacuum-specific impulse. ^e^ density of propellant. ^f^ density-specific impulse of propellant. (These parameters were determined under frozen flow conditions by assuming chamber pressure (P_c_, 2.4 × 10^6^ Pa), exit pressure (P_e_, 9.8 × 10^4^ Pa), and the area ratio (A_e_/A_t_, 4)).

**Table 2 molecules-30-01789-t002:** Physicochemical properties of all ionic liquids based on borohydride-rich anions.

IL	ρ ^a^ (g cm^−3^)	η ^b^ (mPa s)	T_d_ ^c^ (°C)	T_m_ ^d^ (°C)	ΔH_f_ ^e^ (kJ mol^−1^)	I_sp_ ^f^ (s)	O/F ^g^	IDT ^h^ (ms)	State ^i^	Oxidizer
IL-1	-	-	-	-	-	-	-	>3000	liquid	98% H_2_O_2_, 90% H_2_O_2_
IL-2	-	-	-	-	-	-	-	<30	liquid	98% H_2_O_2_, 90% H_2_O_2_
IL-3	0.98	19	247	−71	136	253	3.5	>1000	liquid	95% H_2_O_2_
IL-4	0.95	17	265	<−50	225	253	3.5	>1000	liquid	95% H_2_O_2_
IL-17	0.92	-	130	50	132	258	3.5	18.5	soild	95% H_2_O_2_
IL-18	1.02	58	-	-	-	-	-	28.3	liquid	90% H_2_O_2_
IL-19	1.03	34	-	-	-	-	-	86.8	liquid	90% H_2_O_2_
IL-20	1.01	110	-	-	-	-	-	344	liquid	90% H_2_O_2_
IL-21	1.00	81	-	-	-	-	-	127	liquid	90% H_2_O_2_

^a^ density at 25 °C. ^b^ viscosity at 25 °C. ^c^ decomposition temperature. ^d^ melting temperature. ^e^ heat of formation. ^f^ specific impulse (under frozen flow conditions, P_c_ = 2.4 × 10^6^ Pa, P_e_ = 9.8 × 10^4^ Pa, A_e_/A_t_ = 4). ^g^ oxidizer-to-fuel ratio. ^h^ ignition delay time. ^i^ physical state of the propellant at room temperature.

**Table 3 molecules-30-01789-t003:** Physicochemical properties of all ionic liquids based on thiocyanate anions.

IL	ρ ^a^ (g cm^−3^)	η ^b^ (mPa s)	T_d_ ^c^ (°C)	T_m_ ^d^ (°C)	ΔH_f_ ^e^ (kJ mol^−1^)	I_sp_ ^f^ (s)	O/F ^g^	IDT ^h^ (ms)	State ^i^	Oxidizer
IL-22	1.11	23	-	−6	53	317	3.8	31.7	liquid	96.1% H_2_O_2_
IL-23	1.07	36	-	−29	−5	319	4.2	45	liquid	96.1% H_2_O_2_
IL-24	1.13	28	247	-	−149	313	4.3	26.8	liquid	97.4% H_2_O_2_
IL-25	1.09	87	258	-	−126	317	4.5	33.9	liquid	97.4% H_2_O_2_
IL-26	1.14	38	197	-	93	317	4.2	29.5	liquid	97.4% H_2_O_2_
IL-27	1.07	-	264	55	−162	320	4.5	43.1	solid	97.4% H_2_O_2_
IL-28	1.05	549	261	-	−477	318	4.9	61.9	liquid	97.4% H_2_O_2_
IL-29	1.07	81	232	-	−140	319	4.6	48.9	liquid	97.4% H_2_O_2_
IL-30	1.36	-	219	106	127	309	3.0	7.3	solid	97% H_2_O_2_
IL-31	1.27	-	229	45	58	312	3.4	23.0	solid	97% H_2_O_2_
IL-32	1.14	77	237	2	49	314	3.6	42.8	liquid	97% H_2_O_2_
IL-33	1.09	170	250	−18	11	318	4.0	46.1	liquid	97% H_2_O_2_
IL-34	1.28	-	252	84	−14	310	3.4	16.5	solid	97% H_2_O_2_
IL-35	1.24	-	259	87	−73	312	3.7	20.2	solid	97% H_2_O_2_
IL-36	1.08	126	270	−74	−2	318	4.0	28.4	liquid	97% H_2_O_2_
IL-37	1.26	-	116	66	−176	307	3.0	55.1 (21 °C); 65.1 (1 °C); 83.6 (−25 °C)	solid	97% H_2_O_2_
IL-38	1.13	26	119	0	−170	311	4.1	30.8 (21 °C); 32.8 (1 °C); 51.8 (−25 °C)	liquid	97% H_2_O_2_
IL-39	1.09	34	123	−13	−269	314	4.5	61.8 (21 °C); 69.2 (1 °C); 79.2 (−25 °C)	liquid	97% H_2_O_2_

^a^ density at 25 °C. ^b^ viscosity at 25 °C. ^c^ decomposition temperature. ^d^ melting temperature. ^e^ heat of formation. ^f^ specific impulse (under frozen flow conditions, P_c_ = 1 MPa, A_e_/A_t_ = 330). ^g^ oxidizer-to-fuel ratio. ^h^ ignition delay time. ^i^ physical state of the propellant at room temperature.

**Table 4 molecules-30-01789-t004:** IDTs of different hypergolic fuel (HF) combinations.

Fuels	IL-3 (wt.%)	IL-40 (wt.%)	Furfuryl Alcohol (wt.%)	NaBH_3_CN (wt.%)	ILP-17 (wt.%)	IDT (ms)
HF-1	100	0	0	0	0	>1000
HF-2	95	0	0	0	5	95.5
HF-3	93	0	0	0	7	93.6
HF-4	91	0	0	0	<9	75.5
HF-5	47.5	0	47.5	0	5	80.3
HF-6	46.5	0	46.5	0	7	64.5
HF-7	45.5	0	45.5	0	9	56.0
HF-8	44.5	0	44.5	0	<11	44.5
HF-9	0	100	0	0	0	392.5
HF-10	0	97	0	0	<3	25.3
HF-11	0	54	46	0	10	44.0
HF-12	0	76	19	0	5	82.3
HF-13	0	0	100	0	0	NO
HF-14	0	0	85	10	5	48.8
HF-15	0	0	83	10	7	34.0
HF-16	0	0	81	10	9	30.0
HF-17	0	0	79	10	<11	29.0

**Table 5 molecules-30-01789-t005:** Physicochemical properties of borohydride-rich ionic liquid–H_2_O_2_ propellant systems with promoters.

IL	Promoter	w_ILP_ ^a^ (wt%)	ρ ^b^ (g cm^−3^)	η ^c^ (mPa s)	T_d_ ^d^ (°C)	T_m_ ^e^ (°C)	ΔH_f_ ^f^ (kJ mol^−1^)	I_sp_ ^g^ (s)	IDT ^h^ (ms)	Oxidizer
IL-3	ILP-1	8	3.13	-	-	-	-	237	69	70% H_2_O_2_
									24	95% H_2_O_2_
IL-3	ILP-2	8	2.73	-	-	-	-	237	45	70% H_2_O_2_
									17	95% H_2_O_2_
IL-3	NO	-	0.98	19	247	-	-	269	>1000	95% H_2_O_2_
IL-3	ILP-3	10	1.02	42	220	-	-	265	37	95% H_2_O_2_
IL-3	ILP-4	10	1.02	48	221	-	-	261	36	95% H_2_O_2_
IL-3	ILP-5	10	1.03	50	219	-	-	264	24	95% H_2_O_2_
IL-3	ILP-6	10	1.02	55	214	-	-	262	38	95% H_2_O_2_
IL-40	NO	-	0.93	5	263			267	393	95% H_2_O_2_
IL-40	ILP-3	10	1.00	80	161	-	-	263	30	95% H_2_O_2_
IL-40	ILP-4	10	1.01	87	162	-	-	259	23	95% H_2_O_2_
IL-40	ILP-5	10	1.01	82	160	-	-	262	14	95% H_2_O_2_
IL-40	ILP-6	10	1.01	89	158	-	-	260	28	95% H_2_O_2_
IL-41	NO	-	1.11	35	203	<−70	359	-	NO	90% H_2_O_2_
IL-41	ILP-7	15	-	-	-	-	-	-	ignition	90% H_2_O_2_
IL-42	NO	-	1.13	52	193	<−70	472	-	NO	90% H_2_O_2_
IL-42	ILP-7	15	-	-	-	-	-	-	ignition	90% H_2_O_2_
IL-43	NO	-	1.12	28	241	<−70	242	-	NO	90% H_2_O_2_
IL-43	ILP-7	15	-	-	-	-	-	-	ignition	90% H_2_O_2_
IL-3	ILP-8	5	-	-	-	-	-	-	139	95% H_2_O_2_
IL-3	ILP-9	5	-	-	-	-	-	-	395	95% H_2_O_2_
IL-3	ILP-10	5	-	-	-	-	-	-	887	95% H_2_O_2_
IL-3	ILP-11	5	-	-	-	-	-	-	73	95% H_2_O_2_
IL-3	ILP-12	5	1.30	24	242	-	-	240	87	95% H_2_O_2_
IL-3	ILP-13	10	1.03	28	205	-	-	239	38	95% H_2_O_2_
IL-3	ILP-14	10	1.03	35	210	-	-	240	31	95% H_2_O_2_
IL-5	ILP-13	10	1.00	39	195	-	-	242	89	95% H_2_O_2_
IL-5	ILP-14	10	1.01	47	197	-	-	242	56	95% H_2_O_2_
IL-3	ILP-18	13	0.98	26	156	-	-	240	9.5	95% H_2_O_2_
TG	ILP-18	13	1.05	10	162	-	-	270	9.0	95% H_2_O_2_
IL-3/TG	ILP-18	13	1.02	21	160	-	-	248	7.8	95% H_2_O_2_
ILethCu01 (IL-3/ethanol)	ILP-18	9	0.90	4	-	-	-	317	7.5	95% H_2_O_2_
triglyme	ILP-18	13	1.02	5	156	−45	-	-	18.7	95% H_2_O_2_
TriGIL (IL-3/triglyme)	ILP-18	13	1.02	26	233	<−80	-	-	8.0	95% H_2_O_2_
IL-3	ILP-20	10	-	-	>240	-	-	278	37	90% H_2_O_2_
IL-3	ILP-22	10	-	-	>240	-	-	277	62	90% H_2_O_2_
IL-3	ILP-26	10	-	-	>240	-	-	277	31	90% H_2_O_2_
IL-3	ILP-28	10	-	-	>240	-	-	277	70	90% H_2_O_2_
IL-3	ILP-23	10	0.94	-	250	-	261	287	104	90% H_2_O_2_
IL-3	ILP-25	10	0.95	-	257	-	229	287	105	90% H_2_O_2_
IL-3	ILP-29	10	0.96	-	256	-	243	287	94	90% H_2_O_2_
IL-3	ILP-31	10	0.96	-	249	-	232	286	95	90% H_2_O_2_

^a^ concentration of promoter (ILP). ^b^ density at 25 °C. ^c^ viscosity at 25 °C. ^d^ decomposition temperature. ^e^ melting temperature. ^f^ heat of formation. ^g^ specific impulse (ILP-12: under frozen flow conditions, P_c_ = 25 atm, P_e_ = 1 atm, A_e_/A_t_ = 4; ILP-13 to ILP-14: under equilibrium conditions, P_c_ = 2.5 MPa; ILP-18 with IL-3: under frozen flow conditions, P_c_ = 25 atm, P_e_ = 1 atm, A_e_/A_t_ = 4; ILP-18 with TG, IL-3/TG and ILethCu01: under frozen flow conditions, P_c_ = 1 MPa, A_e_/A_t_ = 100; ILP-18 with triglyme and TriGIL: under frozen flow conditions, P_c_ = 9 bar, A_e_/A_t_ = 150). ^h^ ignition delay time.

**Table 6 molecules-30-01789-t006:** Physicochemical properties of thiocyanate-rich ionic liquid–H_2_O_2_ propellant systems with promoters.

IL	Promoter	w_ILP_ ^a^ (wt%)	ρ ^b^ (g cm^−3^)	η ^c^ (mPa s)	T_d_ ^d^ (°C)	T_m_ ^e^ (°C)	ΔH_f_ ^f^ (kJ mol^−1^)	I_sp_ ^g^ (s)	IDT ^h^ (ms)	Oxidizer
IL-22	ILP-32	5	1.15	30	273	-	-	-	13.9	96.1% H_2_O_2_
IL-22	NO	-	-	-	-	-	-	320	23.0	95% H_2_O_2_
IL-22	ILP-35	0.5	-	-	-	-	-	320	22.9	95% H_2_O_2_
IL-22	ILP-35	1	-	-	-	-	-	319	21.5	95% H_2_O_2_
IL-22	ILP-35	5	-	-	-	-	-	316	19.3	95% H_2_O_2_
IL-22	ILP-35	20	-	-	-	-	-	302	15.0	95% H_2_O_2_
IL-22	ILP-36	0.5	-	-	-	-	-	320	21.6	95% H_2_O_2_
IL-22	ILP-36	1	-	-	-	-	-	320	21.9	95% H_2_O_2_
IL-22	ILP-36	5	-	-	-	-	-	319	23.0	95% H_2_O_2_
IL-22	ILP-36	20	-	-	-	-	-	313	27.8	95% H_2_O_2_
IL-23	NO	-	-	-	-	-	-	322	35.1	95% H_2_O_2_
IL-23	ILP-35	0.5	-	-	-	-	-	322	33.1	95% H_2_O_2_
IL-23	ILP-35	1	-	-	-	-	-	321	32.2	95% H_2_O_2_
IL-23	ILP-35	5	-	-	-	-	-	318	30.1	95% H_2_O_2_
IL-23	ILP-35	20	-	-	-	-	-	303	23.7	95% H_2_O_2_
IL-23	ILP-36	0.5	-	-	-	-	-	320	34.6	95% H_2_O_2_
IL-23	ILP-36	1	-	-	-	-	-	322	32.7	95% H_2_O_2_
IL-23	ILP-36	5	-	-	-	-	-	320	31.6	95% H_2_O_2_
IL-23	ILP-36	20	-	-	-	-	-	314	36.7	95% H_2_O_2_

^a^ concentration of promoter (ILP). ^b^ density at 25 °C. ^c^ viscosity at 25 °C. ^d^ decomposition temperature. ^e^ melting temperature. ^f^ heat of formation. ^g^ specific impulse (under equilibrium conditions, P_c_ = 2 MPa, A_e_/A_t_ = 70). ^h^ ignition delay time.

**Table 7 molecules-30-01789-t007:** Physicochemical properties of dicyanamide-rich ionic liquid–H_2_O_2_ propellant systems with promoters.

IL	Promoter	w_ILP_ ^a^ (wt%)	ρ ^b^ (g cm^−3^)	η ^c^ (mPa s)	T_d_ ^d^ (°C)	T_m_ ^e^ (°C)	ΔH_f_ ^f^ (kJ mol^−1^)	I_sp_ ^g^ (s)	IDT ^h^ (ms)	Oxidizer
IL-50	ILP-37	8	-	-	-	-	-	-	110	90% H_2_O_2_
IL-50	ILP-37	8	-	-	-	-	-	-	130	98% H_2_O_2_
IL-51	ILP	15	-	-	-	−5	-	-	9	95% H_2_O_2_
IL-51	ILP	15	-	-	-	−11	-	-	11	90% H_2_O_2_
IL-51	ILP	15	-	-	-	−25	-	-	22	80% H_2_O_2_
IL-51	ILP	15	-	-	-	−38	-	-	66	70% H_2_O_2_
IL-52	NO	-	1.12	69	205	-	-	235	>1000	95% H_2_O_2_
IL-52	ILP-13	10	1.14	78	182	-	-	235	54	95% H_2_O_2_
IL-52	ILP-14	10	1.15	84	188	-	-	235	42	95% H_2_O_2_
IL-40	ILP-41	5	1.14	17	257	<−70	-	272	75	90% H_2_O_2_
IL-40	ILP-42	3	1.13	19	256	<−70	-	275	112	90% H_2_O_2_
IL-40	ILP-42	5	1.14	17	254	<−70	-	274	74	90% H_2_O_2_
IL-40	ILP-42	10	1.15	24	252	<−70	-	272	59	90% H_2_O_2_
IL-40	ILP-42	15	1.20	31	248	<−70	-	270	47	90% H_2_O_2_
IL-40	ILP-42	20	1.25	40	244	<−70	-	268	34	90% H_2_O_2_
IL-53	ILP-41	5	1.09	27	273	<−70	-	264	161	90% H_2_O_2_
IL-53	ILP-42	5	1.09	27	272	<−70	-	266	113	90% H_2_O_2_
IL-52	ILP-43	12	1.13	35	238	-	-	297	16	90% H_2_O_2_
IL-52	ILP-44	12	1.15	37	240	-	-	297	36	90% H_2_O_2_
IL-52	ILP-45	12	1.15	35	236	-	-	298	29	90% H_2_O_2_
IL-52	ILP-46	12	1.13	34	238	-	-	296	20	90% H_2_O_2_
IL-52	ILP-47	12	1.11	40	239	-	-	299	59	90% H_2_O_2_
IL-51	ILP-43	12	1.13	42	239	-	-	296	26	90% H_2_O_2_
IL-53	ILP-43	12	1.10	52	243	-	-	295	36	90% H_2_O_2_
IL-54	ILP-43	12	1.17	40	212	-	-	298	27	90% H_2_O_2_
IL-55	ILP-43	12	1.16	43	214	-	-	298	17	90% H_2_O_2_

^a^ concentration of promoter (ILP). ^b^ density at 25 °C. ^c^ viscosity at 25 °C. ^d^ decomposition temperature. ^e^ melting temperature. ^f^ heat of formation. ^g^ specific impulse (ILP-13 to ILP-14: under equilibrium conditions, P_c_ = 2.5 MPa; ILP-41 to ILP-42: using EXPLO5 with P_c_ and initial temperature as 7 MPa and 3300 K; ILP-43 to ILP-47: using NASA CEA with P_c_ and ambient pressure as 69.8 × 10^5^ Pa and 1.0 × 10^5^ Pa). ^h^ ignition delay time.

**Table 8 molecules-30-01789-t008:** Physicochemical properties of other ionic liquid–H_2_O_2_ propellant systems with promoters.

IL	Promoter	w_ILP_ ^a^ (wt%)	ρ ^b^ (g cm^−3^)	η ^c^ (mPa s)	T_d_ ^d^ (°C)	T_m_ ^e^ (°C)	ΔH_f_ ^f^ (kJ mol^−1^)	I_sp_ ^g^ (s)	IDT ^h^ (ms)	Oxidizer
IL-56	ILP-37	14	-	-	-	-	-	-	170	98% H_2_O_2_
IL-57	ILP-37	22	-	-	-	-	-	-	50	98% H_2_O_2_
IL-58	ILP-37	20	-	-	-	-	-	-	960	98% H_2_O_2_
IL-59	ILP-48/49	10/8.7	1.05	97	-	-	-	326	28	97% H_2_O_2_
IL-59	ILP-48/49	19.4/7.8	1.02	37	-	-	-	326	28	97% H_2_O_2_

^a^ concentration of promoter (ILP). ^b^ density at 25 °C. ^c^ viscosity at 25 °C. ^d^ decomposition temperature. ^e^ melting temperature. ^f^ heat of formation. ^g^ specific impulse (under frozen flow conditions, P_c_ = 10 bar, A_e_/A_t_ = 330). ^h^ ignition delay time.

## Data Availability

This article is a review of previous works, no new data were created. The related data can be found in the references.

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
