# Peer review of "Advances in Synthesis and Ignition Performance of Ionic Liquid–Hydrogen Peroxide Green Propellants"

_molecules, 2025, doi:10.3390/molecules30081789_

Round 1
Reviewer 1 Report
Comments and Suggestions for Authors
The manuscript titled "Advances in Synthesis and Ignition Performance of Ionic Liquid-Hydrogen Peroxide Green Propellants" reviews the latest research in the field of ionic liquid fuel and hydrogen peroxide oxidizer. The data is well-organized and effectively discussed. The manuscript can be considered for publication after minor revision.
- The author should review the NASA CEA parameters used for Isp calculations. These parameters include chamber pressure, area ratio, and whether the calculations assume frozen or equilibrium conditions. This information can be included in the table footnote.
- Author should discussed the HIP_11 fuel used for hot firing test in section 3.2 (https://doi.org/10.2514/1.B38413)
- Another paper need to discuss about ionic liquid and triglyme fuels (https://doi.org/10.1016/j.combustflame.2024.113719).
- What do the authors think is the best-performing ionic liquid-promoter with hydrogen peroxide for real rocket applications.
- How do viscosity, cost, and stability of the ionic liquid-promoter affect the development of green hypergolic fuels?
Author Response
Responses to the reviewer 1#:
Comments: The manuscript titled "Advances in Synthesis and Ignition Performance of Ionic Liquid-Hydrogen Peroxide Green Propellants" reviews the latest research in the field of ionic liquid fuel and hydrogen peroxide oxidizer. The data is well-organized and effectively discussed. The manuscript can be considered for publication after minor revision.
Response: Thanks for the reviewer’s recognition and careful inspection on our work. We have carefully studied the comments and have made the corrections as below. We hope that these answers and corrections will meet with the approval.
-------------------------------------------------------------------------------------------------------
Question 1: The author should review the NASA CEA parameters used for Isp calculations. These parameters include chamber pressure, area ratio, and whether the calculations assume frozen or equilibrium conditions. This information can be included in the table footnote.
Response: We sincerely appreciate the reviewer’s insightful comments and constructive feedback. We recognize the importance of clearly specifying the NASA CEA parameters used for Isp calculations. To address this concern, we have carefully reviewed and revised the relevant sections of the manuscript. Specifically, we have incorporated the relevant details into the footnotes of the tables in the revised manuscript, as outlined below:
(1) Page 3, line 110, “(These parameters were determined under frozen flow conditions by assuming chamber pressure (Pc, 2.4 × 106 Pa), exit pressure (Pe, 9.8 × 104 Pa) and the area ratio (Ae/At, 4).)”;
(2) Page 5, line 163, “(under frozen flow conditions, Pc = 2.4 × 106 Pa, Pe = 9.8 × 104 Pa, Ae/At = 4)”;
(3) Page 8, line 241, “(under frozen flow conditions, Pc = 1 MPa, Ae/At = 330)”;
(4) Page 15, line 421, “(ILP-12: under frozen flow conditions, Pc = 25 atm, Pe = 1 atm, Ae/At = 4; ILP-13 to ILP-14: under equilibrium conditions, Pc = 2.5 MPa; ILP-18 with IL-3: under frozen flow conditions, Pc = 25 atm, Pe = 1 atm, Ae/At = 4; ILP-18 with TG, IL-3/TG and ILethCu01: under frozen flow conditions, Pc = 1 MPa, Ae/At = 100; ILP-18 with triglyme and TriGIL: under frozen flow conditions, Pc = 9 bar, Ae/At = 150)”;
(5) Page 17, line 503, “(under equilibrium conditions, Pc = 2 MPa, Ae/At = 70)”;
(6) Page 21, line 602, “(ILP-13 to ILP-14: under equilibrium conditions, Pc = 2.5 MPa; ILP-41 to ILP-42: using EXPLO5 with Pc and initial temperature as 7 MPa and 3300 K; ILP-43 to ILP-47: using NASA CEA with Pc and ambient pressure as 69.8 × 105 Pa and 1.0 × 105 Pa)”;
(7) Page 21, line 637, “(under frozen flow conditions, Pc = 10 bar, Ae/At = 330)”.
-------------------------------------------------------------------------------------------------------
Question 2: Author should discussed the HIP_11 fuel used for hot firing test in section 3.2 (https://doi.org/10.2514/1.B38413).
Response: We sincerely appreciate your careful review and thoughtful suggestion. Your insights have been invaluable in improving the clarity and completeness of our manuscript. We have added a detailed statement about the HIP_11 fuel on line 448 of page 16: “In 2022, Lauck et al. investigated the performance of a novel green hypergolic propellant, HIP_11, composed of 97% H2O2 as the oxidizer and a fuel blend of IL-22 with 5% ILP-32, aiming to evaluate the ignition reliability, combustion efficiency, and stability in a small "battleship" thruster designed for robustness and repeated testing[36]. Initial injector tests demonstrated that increasing the ILP-32 concentration from 1% to 5% markedly reduced IDTs to 13.9 ms, ensuring consistent hypergolic initiation. Subsequent hot-fire campaigns achieved stable combustion at chamber pressures exceeding 7 bar, delivering 93–98% combustion efficiency with exceptionally low pressure oscillations (1.1–1.7% of chamber pressure) and rapid rise times under 13 ms. This study marked the first successful demonstration of an ionic liquid/hydrogen peroxide system in operational thruster conditions, showing HIP_11’s viability as a low-toxicity alternative to conventional propellants. The system’s simplicity, repeatable ignition performance, and efficiency comparable to established hypergolic technologies highlight its potential for practical aerospace applications while addressing environmental and safety concerns.”
-------------------------------------------------------------------------------------------------------
Question 3: Another paper need to discuss about ionic liquid and triglyme fuels (https://doi.org/10.1016/j.combustflame.2024.113719).
Response: We sincerely appreciate your insightful suggestion and careful review. We have carefully considered your suggestion and incorporated the necessary details on line 358 of page 12: “In the same year, Seo et al. investigated the development of low-toxicity hypergolic propellants by enhancing triglyme-based fuels with ILP-18 and IL-3[32]. Their study evaluated the IDTs of triglyme and triglyme-ionic liquid blends (TriGIL) containing varying ILP-18 concentrations, tested with 70, 90, and 95% H2O2. The 13Cu-TriGIL(TriGIL with 13% ILP-18) demonstrated exceptional performance, achieving an 8.0 ms IDT with 95% H2O2 compared to 18.7 ms for triglyme alone. The ionic liquid integration also improved cryogenic stability, suppressing the blend’s freezing point from -45 ℃ (pure triglyme) to below -80 ℃. The optimized fuel exhibited favorable propulsion characteristics, including a density of 1.017 g cm-3 and viscosity of 26.42 mPa s. Theoretical calculations revealed that 13Cu-TriGIL with 95% H2O2 exhibited a 9.7% higher ρIsp than MMH/NTO, despite a 2.1% lower Isp. This work underscores triglyme-ionic liquid blends as promising eco-friendly alternatives to toxic hypergolic propellants, combining rapid ignition, enhanced thermal stability, and reduced hazards while maintaining competitive performance metrics.”
-------------------------------------------------------------------------------------------------------
Question 4: What do the authors think is the best-performing ionic liquid-promoter with hydrogen peroxide for real rocket applications.
Response:. We thank the reviewer for the insightful question and professional suggestion. We have carefully considered and incorporated a detailed discussion on line 668 of page 22: “Among the promoters analyzed, ILP-18 emerges as the most promising candidate for rocket propulsion. As shown in Table 5, the IL-3/ethanol blend (ILethCu01) with 9% ILP-18 achieves an ultralow IDT of 7.5 ms, a low viscosity of 3.85 mPa s, and high Isp of 317 s. Notably, ILP-18 retains reliable hypergolic ignition even at extreme temperatures (the TriGIL system can operate at −80 ℃). This catalyst functions through two complementary mechanisms: copper ions accelerate the decomposition of hydrogen peroxide, while borohydrides drive rapid redox reactions. These processes enable fast ignition and sustained combustion, meeting the practical requirements of rocket engines. These attributes position ILP-18 as the leading green propellant candidate for next-generation propulsion systems.”
-------------------------------------------------------------------------------------------------------
Question 5: How do viscosity, cost, and stability of the ionic liquid-promoter affect the development of green hypergolic fuels?
Response: We sincerely appreciate your careful review and thoughtful suggestion. These parameters of the ionic liquid-promoter are crucial roles in fuel formulation design. We have carefully considered your question and incorporated the necessary details on line 646 of page 22: “The viscosity, Isp, melting point, stability, solubility and cost of promoters are critical determinants of their practical applicability. First, viscosity directly influences propellant fluidity, which governs injection efficiency and combustion performance. For example, adding ILP-32 to IL-22 elevates viscosity from 20 to 29.6 mPa s, while blending IL-52 with ILP-13 or ILP-14 increases viscosity from 69 to 78 and 84 mPa s, respectively. Elevated viscosity raises flow resistance, impairing injection efficiency, and poor mixing due to high viscosity can lead to incomplete combustion. To address this, solvent blending or the design of low-viscosity promoters (η < 50 mPa s) is essential. Isp is an important index to measure the performance of propellant. Higher Isp (> 300 s) can enhance rocket engine efficiency, payload capacity and velocity. Low melting points (Tm < room temperature) ensure propellant liquidity under cryogenic conditions, preventing solidification and ensuring reliable supply. Thermal stability (Td > 200 ℃) is vital for safe storage and consistent ignition in engines. The solubility of solid promoters is also a critical parameter in fuel formulation design. These promoters must be uniformly dispersed within the fuel to form a homogeneous system, ensuring sufficient contact with the fuel and enabling effective catalytic action. However, low solubility can lead to phase separation or sedimentation, resulting in uneven local concentrations. This imbalance may cause delayed or incomplete combustion, ultimately reducing specific impulse and compromising reliability. Cost considerations are equally critical: metal-containing promoters like ILP-41 offer superior catalytic efficiency but require complex synthesis and expensive metals, whereas metal-free alternatives like ILP-32 are cost-effective but less active. Future advancements must balance catalytic performance, economic viability, and environmental sustainability.”
Reviewer 2 Report
Comments and Suggestions for Authors
This manuscript reviewed the researches of the ionic liquid-hydrogen peroxide propellant system, which has extensive and detailed content about this kind of liquid propellant. It also addressed the current challenges and future directions in this field. Obviously, it has an important reference value for researchers studying energetic ionic liquid propellant. However, there are still some areas that need further improvement. A major revision is recommended. Specific suggestions are as follows:
- The ignition delay time data in the manuscript should be indicated which values are experimental and which are calculated to provide better reference for readers.
- Could the authors establish a criterion and use it to highlight the most promising energetic liquid fuel formulations in the tables? This would offer readers a more direct reference.
- The manuscript presents numerous bipropellants and multi-component liquid propellant formulations but only describes them briefly. It is recommended that a summary analysis at the end of each section should be given to evaluate the research findings on these propellants, making the review more valuable.
- It is suggested to add a column in the tables indicating the physical state of the propellant at room temperature (liquid or solid), which would help readers quickly assess the overall performance of the formulations.
Author Response
Response to Reviewer 2#:
Comments: This manuscript reviewed the researches of the ionic liquid-hydrogen peroxide propellant system, which has extensive and detailed content about this kind of liquid propellant. It also addressed the current challenges and future directions in this field. Obviously, it has an important reference value for researchers studying energetic ionic liquid propellant. However, there are still some areas that need further improvement. A major revision is recommended. Specific suggestions are as follows:
Response: We sincerely appreciate the reviewers' recognition and careful inspection of our work. We have carefully studied each comment and implemented the suggested revisions. We hope that these answers and corrections will meet with the approval.
-------------------------------------------------------------------------------------------------------
Question 1: The ignition delay time data in the manuscript should be indicated which values are experimental and which are calculated to provide better reference for readers.
Response: We sincerely appreciate your insightful suggestion and careful review. We recognize the importance of clearly distinguishing between experimental and calculated data to provide readers with accurate references. In response to your comment, we would like to clarify that all the ignition delay time data presented in our manuscript are derived from experimental results. While Table 1 (Page 3, line 106) does include calculations of specific impulse and other theoretical properties for some ionic liquids, it does not involve any predictions of ignition delay time. Therefore, we have made an explanation on line 105 of page 3 in the revised manuscript: “Additionally, unless specifically noted, all IDT data presented in this study were obtained through experimental measurements.” We hope this clarification addresses your concern and enhances the clarity of our presentation. Thank you again for your guidance and support in improving our research.
-------------------------------------------------------------------------------------------------------Question 2: Could the authors establish a criterion and use it to highlight the most promising energetic liquid fuel formulations in the tables? This would offer readers a more direct reference.
Response: We thank the reviewer for the professional suggestions. We have carefully examined and summarized the effects of the various performance parameters of the propellant on line 646 of page 22: “The viscosity, Isp, melting point, stability, solubility and cost of promoters are critical determinants of their practical applicability. First, viscosity directly influences the propellant fluidity, which governs injection efficiency and combustion performance. For example, adding ILP-32 to IL-22 elevates the viscosity from 20 to 29.6 mPa s, while blending IL-52 with ILP-13 or ILP-14 increases the viscosity from 69 to 78 and 84 mPa s, respectively. Elevated viscosity raises flow resistance, impairing injection efficiency, and poor mixing due to high viscosity can lead to incomplete combustion. To address this, solvent blending or the design of low-viscosity promoters (η < 50 mPa s) is essential. Isp is an important index to measure the performance of propellant. Higher Isp (> 300 s) can enhance rocket engine efficiency, payload capacity and velocity. Low melting points (Tm < room temperature) ensure propellant liquidity under cryogenic conditions, preventing solidification and ensuring reliable supply. Thermal stability (Td > 200 ℃) is vital for safe storage and consistent ignition in engines. The solubility of solid promoters is also a critical parameter in fuel formulation design. These promoters must be uniformly dispersed within the fuel to form a homogeneous system, ensuring sufficient contact with the fuel and enabling effective catalytic action. However, low solubility can lead to phase separation or sedimentation, resulting in uneven local concentrations. This imbalance may cause delayed or incomplete combustion, ultimately reducing specific impulse and compromising reliability. Cost considerations are equally critical: metal-containing promoters like ILP-41 offer superior catalytic efficiency but require complex synthesis and expensive metals, whereas metal-free alternatives like ILP-32 are cost-effective but less active. Future advancements must balance catalytic performance, economic viability, and environmental sustainability.”
Based on the results of the above analysis and discussion, we summarize what we consider to be the most promising propellant formulation and describe it in detail on line 668 of page 22: “Among the promoters analyzed, ILP-18 emerges as the most promising candidate for rocket propulsion. As shown in Table 5, the IL-3/ethanol blend (ILethCu01) with 9% ILP-18 achieves an ultralow IDT of 7.5 ms, a low viscosity of 3.85 mPa s, and high Isp of 317 s. Notably, ILP-18 retains reliable hypergolic ignition even at extreme temperatures (the TriGIL system can operate at −80 ℃). This catalyst functions through two complementary mechanisms: copper ions accelerate the decomposition of hydrogen peroxide, while borohydrides drive rapid redox reactions. These processes enable fast ignition and sustained combustion, meeting the practical requirements of rocket engines. These attributes position ILP-18 as the leading green propellant candidate for next-generation propulsion systems.”
We hope these additions enhance the clarity and completeness of our presentation and align with the reviewer’s expectations. Thank you again for your guidance in improving the quality of our work.
------------------------------------------------------------------------------------------------------- Question 3: The manuscript presents numerous bipropellants and multi-component liquid propellant formulations but only describes them briefly. It is recommended that a summary analysis at the end of each section should be given to evaluate the research findings on these propellants, making the review more valuable.
Response: We sincerely thank the reviewer’s insightful observation and constructive feedback. While the original manuscript included section summaries, we recognize that our initial efforts fell short of fully synthesizing and comparing these findings. We sincerely appreciate the reviewer’s guidance in highlighting this area for improvement, and we have carefully restructured and modified the section summaries to better address this critical aspect. The specific descriptions added are as follows:
(1) Section 2.1, page 5, line 155, “IL-18 to IL-21 had high densities and self-ignition behavior owing to the solvent (DBN and DBU), which were the main component of the fuels. These limitations highlight two research priorities for borohydride-rich anion architectures: experimentally vali-dating computationally predicted hypergolic candidates, and developing novel ionic liquids that combine room-temperature liquidity, solvent-free formulations, and spontaneous ignition capabilities.”
(2) Section 2.2, page 7, line 222, “The physicochemical properties of all ionic liquid based on thiocyanate anions are shown in Table 3. These ionic liquids exhibit rapid hypergolic ignition with high-concentration H2O2, achieving a minimum IDT of 7.3 ms and a high Isp of up to 320 s. Cationic structural design plays a critical role in their performance. Shorter alkyl chains on pyridinium, pyrrolidinium, and imidazolium cations significantly reduce IDTs. However, these systems are heavily dependent on the high-concentration H2O2, and lower temperatures markedly degrade the ignition efficiency (e.g., IL-37’s IDT increases to 83.6 ms at -25 ℃). Notably, trialkylsulfonium-based IL-38 defies the alkyl chain length trend, achieving the shortest IDT of 30.8 ms despite longer substituents, suggesting potential asymmetry effects in cationic frameworks. Future investigations should prioritize these objectives: the first is elucidating the structure-ignition relationship in trialkylsulfonium cations to clarify the role of asymmetry, and the second is expanding the library of hypergolic thiocyanate-based ionic liquids through systematic cation-anion pairing strategies. Additionally, current systems require high-concentration H2O2 for spontaneous ignition, so further research is essential to evaluate their compatibility with low-concentration H2O2 which could enhance practicality and environmental adaptability.”
(3) Section 3.1, page 14, line 415, “Meanwhile, increased viscosity may impair combustion efficiency. The viscosity of IL-5 with ILP-14 rises from 28 to 47 mPa s. Future efforts should prioritize designing multifunc-tional promoters that balance catalytic activity with physicochemical properties, alongside eco-friendly and cost-effective synthesis routes.”
(4) Section 3.2, page 16, line 494, “Notably, ILP-32 at 5% concentration achieved the shortest IDT of 13.9 ms, outperforming the other two nitrates ILP-35 and ILP-36. Further analysis suggests that ILP-35 exhibits superior catalytic performance compared to ILP-36, likely due to its specific metal ion content, which accelerates reaction kinetics. These findings underscore the importance of continued exploration of inorganic salt promoters to optimize hypergolic performance in bisulfate ionic liquid-hydrogen peroxide systems, highlighting inorganic additives as a priority for future research.”
(5) Section 3.3, page 19, line 592, “achieving IDTs below 40 ms. Despite these advancements, inherent trade-offs persist between catalytic activity and physicochemical properties, necessitating continued innovation.”
(6) Section 3.4, page 21, line 628, “Notably, metal-ion promoters (such as ILP-37) have been extended to non-conventional ionic liquids (such as IL-56/57/58), but their performance exhibits significant variability. For instance, IDTs range from 50 ms for IL-57 to 960 ms for IL-58, underscoring the critical role of ionic liquid selection in system optimization. Solvent blending strategies, exemplified by combinations like IL-59+ILP-48/49, effectively reduce viscosity (from 97 to 37 mPa s), thereby improving fluidity.”
-------------------------------------------------------------------------------------------------------
Question 4: It is suggested to add a column in the tables indicating the physical state of the propellant at room temperature (liquid or solid), which would help readers quickly assess the overall performance of the formulations.
Response: We sincerely appreciate the reviewer’s valuable suggestion regarding the indication of propellants’ physical state at room temperature. Following your insightful comment, the related descriptions in Section 2 were added to emphasize the physical state of the ionic liquid at room temperature, as highlighted in Table 2 (page 5, line 161) and Table 3 (page 7, line 239).
Regarding the composite fuels discussed in Section 3 (consisting of ionic liquid and promoter), we recognize the complexity of defining a unified physical state for these mixtures. Through careful consideration, we found that the solubility of promoter in ionic liquid serves as a more precise indicator for assessing fuel performance. Accordingly, we have thoroughly analyzed the influence of promoter solubility on propellant characteristics (page 22, line 658) and systematically integrated this critical factor into our evaluation framework. The specific description is as follows: “The solubility of solid promoters is also a critical parameter in fuel formulation design. These promoters must be uniformly dispersed within the fuel to form a homogeneous system, ensuring sufficient contact with the fuel and enabling effective catalytic action. However, low solubility can lead to phase separation or sedimentation, resulting in uneven local concentrations. This imbalance may cause delayed or incomplete combustion, ultimately reducing specific impulse and compromising reliability.”
Round 2
Reviewer 2 Report
Comments and Suggestions for Authors
No more questions. Obviously, the manuscript has been revised speicifically. An accept is recommended.